# Boosting NAD+ with a small molecule that activates NAMPT

Stephen J. Gardell[1,2], Meghan Hopf[1,2], Asima Khan[1,2], Mauro Dispagna[1], E. Hampton Sessions[3], Rebecca Falter[3], Nidhi Kapoor[2,4], Jeanne Brooks[4], Jeffrey Culver[4], Chris Petucci[4], Chen-Ting Ma[5], Steven E. Cohen [6], Jun Tanaka [7], Emmanuel S. Burgos [8], Jennifer S. Hirschi[9], Steven R. Smith[1,2], Eduard Sergienko[5] & Anthony B. Pinkerton[5]

Pharmacological strategies that boost intracellular NAD+ are highly coveted for their therapeutic potential. One approach is activation of nicotinamide phosphoribosyltransferase (NAMPT) to increase production of nicotinamide mononucleotide (NMN), the predominant NAD+ precursor in mammalian cells. A high-throughput screen for NAMPT activators and hit-to-lead campaign yielded SBI-797812, a compound that is structurally similar to active-site directed NAMPT inhibitors and blocks binding of these inhibitors to NAMPT. SBI-797812 shifts the NAMPT reaction equilibrium towards NMN formation, increases NAMPT affinity for ATP, stabilizes phosphorylated NAMPT at His247, promotes consumption of the pyrophosphate by-product, and blunts feedback inhibition by NAD+. These effects of SBI-797812 turn NAMPT into a "super catalyst" that more efficiently generates NMN. Treatment of cultured cells with SBI-797812 increases intracellular NMN and NAD+. Dosing of mice with SBI-797812 elevates liver NAD+. Small molecule NAMPT activators such as SBI-797812 are a pioneering approach to raise intracellular NAD+ and realize its associated salutary effects.

[1] Center for Metabolic Origins of Disease, Sanford Burnham Prebys Medical Discovery Institute at Lake Nona, Orlando, FL 32827, USA. [2] Translational Research Institute for Metabolism and Diabetes, AdventHealth-Orlando, Orlando, FL 32804, USA. [3] Conrad Prebys Center for Chemical Genomics, Sanford Burnham Prebys Medical Discovery Institute at Lake Nona, Orlando, FL 32827, USA. [4] Metabolomics Core Facility, Sanford Burnham Prebys Medical Discovery Institute at Lake Nona, Orlando, FL 32827, USA. [5] Conrad Prebys Center for Chemical Genomics, Sanford Burnham Prebys Medical Discovery Institute, La Jolla, CA 92037, USA. [6] Daiichi Sankyo, Inc., Global Business Development, Basking Ridge, NJ 07920, USA. [7] Daiichi Sankyo Co., Ltd, Shinagawa Research & Development Center, Tokyo 140-8710, Japan. [8] Department of Biochemistry, Albert Einstein College of Medicine, Bronx, NY 10461, USA. [9] Department of Chemistry, Binghamton University, Binghamton, NY 13902, USA. Correspondence and requests for materials should be addressed to S.J.G. (email: stephen.gardell@adventhealth.com) or to A.B.P. (email: apinkerton@sbpdiscovery.org)

NAD[+] plays a vital role in diverse cellular processes that govern human health and disease[1]. The long-standing focus on NAD[+] as a redox enzyme cofactor has been eclipsed by recent seminal discoveries establishing NAD[+] as a co-substrate for sirtuins and poly-ADP-ribose polymerases (PARPs)[2,3]. These revelations have implicated NAD[+] in additional cellular processes including cell signaling, DNA repair, cell division, and epigenetics. Elevated tissue levels of NAD[+] were linked to salutary effects including healthy aging[4]. Thus, there is keen interest in pharmacological and nutraceutical strategies to boost intracellular NAD[+] levels[5,6].

Enzymatic activities catalyzed by sirtuins and PARPs consume intracellular NAD[+][7]. Hence, a cellular biosynthetic pathway to preserve the NAD[+] level is imperative. In mammalian cells, the principle contributor to NAD[+] synthesis is the nicotinamide (NAM) salvage pathway involving sequential actions of nicotinamide phosphoribosyltransferase (NAMPT) and NMN adenylyltransferases (NMNAT1-3)[8]. NAMPT forms NMN and pyrophosphate (PP) from NAM (generated by sirtuins and PARPs) and α-D-5-phosphoribosyl-1-pyrophosphate (PRPP). In turn, NMNAT1-3 produce NAD[+] from NMN and ATP.

NAMPT, a homodimeric type II phosphoribosyltransferase, is the putative rate-limiting step in the NAM salvage pathway[9]. The canonical NAMPT reaction scheme involves the following sequential steps: (1) ATP binding, NAMPT phosphorylation at His247 to form pHisNAMPT, followed by ADP release; (2) PRPP binding to pHisNAMPT, followed by NAM binding and (3) catalysis producing NMN and PP, followed by product release and regeneration of non-phosphorylated NAMPT[10–12]. The NAMPT protein structure with and without various ligands has been solved by X-ray crystallography[10,13–16].

Our pursuit of a pharmacological approach to boost intracellular NAD[+] levels focuses on discovering compounds that increase the activity of NAMPT. High-throughput screening (HTS) of a small molecule library using a protein thermal shift (PTS) assay[17] yielded novel NAMPT ligands. Subsequent evaluation with a NAMPT activity assay identified a subset of HTS hits that increase NMN production. The ensuing medicinal chemistry campaign produced SBI-797812, our NAMPT activator prototype. Herein, we describe the mechanism of action (MOA) of SBI-797812 and test the ability of this small molecule to raise NMN and NAD[+] in cultured cells and mice.

## Results

**Discovery of a small molecule NAMPT activator.** A chemical library (57,004 compounds) was screened for small molecules that bound to human NAMPT using a PTS assay (Fig. 1a). The negative control was DMSO-treated NAMPT. The positive control was NAMPT treated with 20 μM CHS-828 (Fig. 1b), a potent NAMPT inhibitor[18]. NAMPT ligands stabilized the enzyme against thermal denaturation (i.e., increased Tm) as detected by binding of Sypro orange fluorescent dye. Five hundred fifteen compounds (0.9%) were identified as NAMPT ligands. While the majority of the hits from the PTS assay were inhibitors or had no activity, 30 compounds (5.8%) were NAMPT activators as determined with NAMPT activity assays (NAD/NADH-Glo assay kit and NMN fluorometric assay[19]).

The HTS hit, which was the focus of our hit-to-lead campaign, SBI-136892 (Fig. 1b), dose-dependently increased the NAMPT Tm (Supplementary Fig. 1a) and stimulated NAMPT-mediated NMN production (Supplementary Fig. 1b). Interestingly, SBI-136892 was structurally similar to active-site-directed NAMPT inhibitors (possessing a urea core and pyridyl group) such as compound 50 (GNI-50; $IC_{50} = 7$ nM), from Genentech[20] (Fig. 1b). We confirmed that GNI-50 was a potent NAMPT

inhibitor (Fig. 1c). A close structural analog of GNI-50, GNI-5 (Fig. 1b), was shown to bind to the NAMPT active site similarly to FK-866 (Fig. 1b), the prototypical NAMPT inhibitor[20]. Surmising that the 4-pyridyl group was crucial for NAMPT activation, we synthesized the 4-pyridyl analog of GNI-50 to produce SBI-797812 (Fig. 1b). Remarkably, moving the pyridine nitrogen from the 3 position (GNI-50) to 4 position (SBI-797812) converted a potent NAMPT inhibitor to a NAMPT activator. SBI-797812 increased NAMPT-catalyzed NMN synthesis by 2.1-fold. (Fig. 1c). The SBI-797812 isomer with the 2-pyridyl group, SBI-796950 (Fig. 1b), slightly inhibited NMN production (Fig. 1c).

SBI-797812 caused concentration-dependent activation of human NAMPT-mediated NMN production in the presence of NAM, PRPP and ATP (Fig. 1d). After adjusting for baseline NMN synthesis without NAMPT activator, the $EC_{50}$ for SBI-797812 was $0.37 \pm 0.06$ μM and maximal NAMPT activity was $55 \pm 3$ U nmol$^{-1}$. The maximal fold stimulation of NMN formation by SBI-797812 was 2.1-fold. The ability of SBI-797812 to activate NAMPT was abolished by NAMPT inhibitors including GNI-50, FK-866, and CHS-828 (Supplementary Fig. 2).

The close structural similarity between SBI-797812 and NAMPT competitive inhibitors strongly suggested that SBI-797812 also bound to the NAMPT active site. This inference was supported by two independent experimental findings. Firstly, the NAMPT(G217R) mutant that was resistant to inhibition by CHS-828[18] or FK-866 (Fig. 1e) was also refractory to the stimulatory effect of SBI-797812 (Fig. 1e). Modeling showed that the R217 side chain sterically obstructed the pyridinium binding pocket thus precluding inhibitor binding at the NAMPT active site[18]. We inferred that the R217 side chain also blocked SBI-797812 binding to the NAMPT active site. Secondly, FK-866 and CHS-828 prevented direct binding of SBI-797812 to NAMPT (Fig. 1f). This approach used a spin (desalting) column to separate free SBI-797812 from the NAMPT•SBI-797812 complex, and subsequent detection of NAMPT-bound SBI-797812 in the column eluent by mass spectrometry. When SBI-797812 was applied to the spin column, it did not appear in the column eluent. When NAMPT and SBI-797812 were mixed and applied to the column, SBI-797812 was detected in the column eluent along with NAMPT. When the NAMPT and SBI-797812 mixture contained an active site-directed NAMPT inhibitor (FK-866 or CHS-866), SBI-797812 was not found in the eluent.

An earlier compound, P7C3, was claimed to be a direct NAMPT activator[21]. Our investigation of P7C3 using the assays described herein failed to confirm that this compound bound to NAMPT or directly stimulated its activity (Supplementary Fig. 3).

**Mechanistic studies.** ATP promotes NAMPT-mediated NMN formation, but ATP is not obligatory for this reaction[11] (result replicated in Fig. 2a). Notably, stimulation of NMN production by SBI-797812 required ATP (Fig. 2a). SBI-797812 slightly inhibited NMN formation in the absence of ATP. We next examined the impact of SBI-797812 on the affinity of NAMPT for ATP in the presence of NAM and PRPP (Fig. 2b). The $K_m$ values of NAMPT for ATP without and with SBI-797812 were $1.73 \pm 0.32$ and $0.29 \pm 0.03$ mM, respectively. The approximate six-fold lower $K_m$ value of NAMPT for ATP in the presence of SBI-797812 will enable saturation of the enzyme at lower ATP concentrations. The corresponding $V_{max}$ values in the absence and presence of SBI-797812 were $271 \pm 28$ and $316 \pm 9$ U nmol$^{-1}$, respectively.

Crucial insight into the NAMPT activator MOA was gleaned from a reaction equilibrium study (Fig. 2c). NAMPT was incubated with substrates for both the forward (NAM, PRPP) and reverse (NMN, PP) reactions. A time-dependent increase of

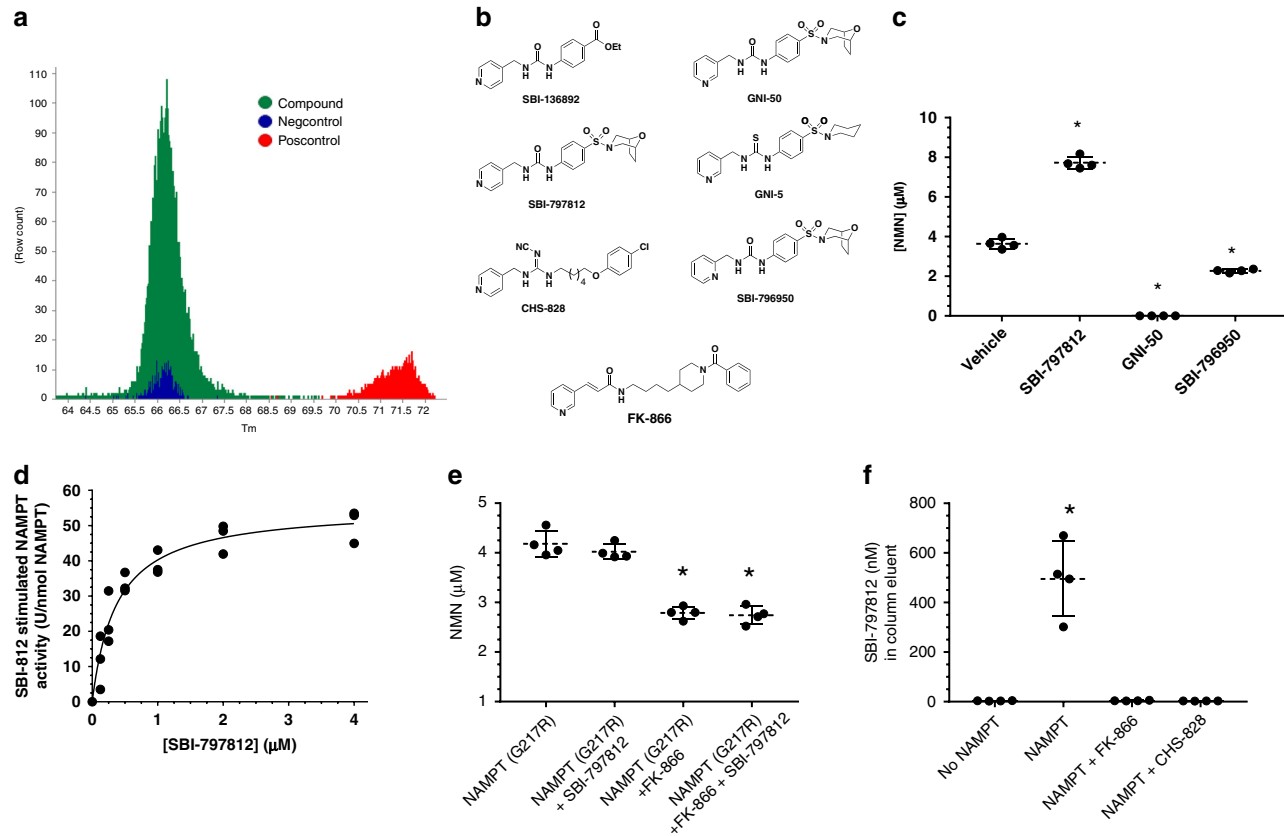

**Fig. 1** Discovery of SBI-797812, a small molecule NAMPT activator. **a** Compiled melting temperature (Tm) data for human NAMPT treated with chemical compounds ($N = 57{,}004$) and tested with the PTS assay protocol (green). NAMPT ligands produced an upward Tm shift. DMSO was the Neg Control (blue). CHS-828 (20 μM) was the Pos Control (red). **b** Structures of NAMPT activators and inhibitors. **c** Pivotal role of the 4-pyridyl nitrogen in SBI-797812 for NAMPT activation. NAMPT (30 nM), NAM (10 μM), PRPP (50 μM), ATP (2 mM) were incubated 1 h at 37 °C with vehicle or 2 μM SBI-797812, GNI-50, or SBI-796950. NMN was detected using the fluorescence assay. Data are expressed as means ± s.d.; $n = 4$. *, $p < 0.0001$ compared to Vehicle. One-way ANOVA with Dunnett's multiple comparisons test was used. **d** Dose-dependent activation of human NAMPT by SBI-797812. NMN production assay was performed as above but with 25 μM NAM and varying SBI-797812. NMN levels were normalized for basal NMN production without SBI-797812. Three replicates were run for each SBI-797812 concentration. Michaelis-Menten curve fit was produced with GraphPad Prism software. **e** NAMPT(G217R) mutant was resistant to both SBI-797812 and FK-866. NAMPT(G217R) (50 nM) was incubated with NAM (10 μM), PRPP (50 μM), ATP (2 mM) and (where indicated) 1 μM SBI-797812 and/or 1 μM FK-866. Reactions were performed for 1 h at 37 °C. NMN was detected with the fluorescence assay. Data are expressed as means ± s.d.; $n = 4$. *$p < 0.0001$ compared to NAMPT(G217R). One-way ANOVA with Dunnett's multiple comparisons test was used. **f** FK-866 and CHS-828 blocked binding of SBI-797812 to NAMPT. SBI-797812 was added to T8MD-Tween buffer with ATP or the same buffer containing NAMPT, NAMPT + FK-866, or NAMPT + CHS-828. Samples were incubated at 37 °C for 10 min and applied to a spin column to separate NAMPT-bound SBI-797812 from unbound SBI-797812. SBI-797812 (in column eluent) was measured by LC-MS-TOF. Data are expressed as means ± s.d.; $n = 4$. *$p < 0.0001$ compared to "No NAMPT". One-way ANOVA with Dunnett's multiple comparisons test was used. For Fig. 1c–f, source data are provided as a Source Data file

NAM and concomitant decrease of NMN was observed during the first 3 h. The dominance of the reverse reaction leading to nearly complete conversion of NMN to NAM was shown previously[11]. Under these same assay conditions, SBI-797812 had no discernible impact on the reaction. After 3 h, ATP was added, and the reactions proceeded for an additional 2 h. Without SBI-797812, a time-dependent NMN increase and concomitant NAM depletion was recorded. Hence, ATP caused the directionality of NAMPT activity to pivot towards the forward reaction. This ATP effect was also consistent with published data[11]. At equilibrium, the NAM:NMN ratio was approximately 6.5:3.5. Notably, SBI-797812 elicited a further dramatic shift of the reaction equilibrium towards NMN synthesis. NAMPT converted nearly all of the NAM to NMN within 1 h after ATP addition in the presence of SBI-797812.

The NAMPT forward reaction (NAM + PRRP + ATP ± SBI-797812) was performed and the levels of NAM, NMN, PP, ADP, and Pi were measured (Fig. 2d). NAM consumption and NMN production were directly correlated, and both reactions were markedly stimulated by SBI-797812 (thus corroborating the results shown in Fig. 2c). The synchronous ATPase activity of NAMPT produced equimolar amounts of ADP and Pi in a reaction that was also stimulated by SBI-797812. The ratio of ADP to NMN production in the absence and presence of SBI-797812 were comparable (1.9- and 1.8-fold, respectively) (Supplementary Fig. 4). Surprisingly, PP accumulation was not stoichiometric with NMN accumulation. This PP shortfall was more prominent in the presence of SBI-797812 (Fig. 2d).

To shed light on the anomalous PP production, NAMPT was incubated with PP or (ATP + PP) and assayed for residual PP. NAMPT did not deplete PP in the absence of ATP (Fig. 3a). In contrast, incubation of NAMPT with PP and ATP resulted in a diminished PP level. Such PP consumption was markedly enhanced by SBI-797812 (Fig. 3a) and abolished by the NAMPT inhibitor, CHS-828 (Supplementary Fig. 5). The fate of the PP was predicted by modeling using density functional theory to

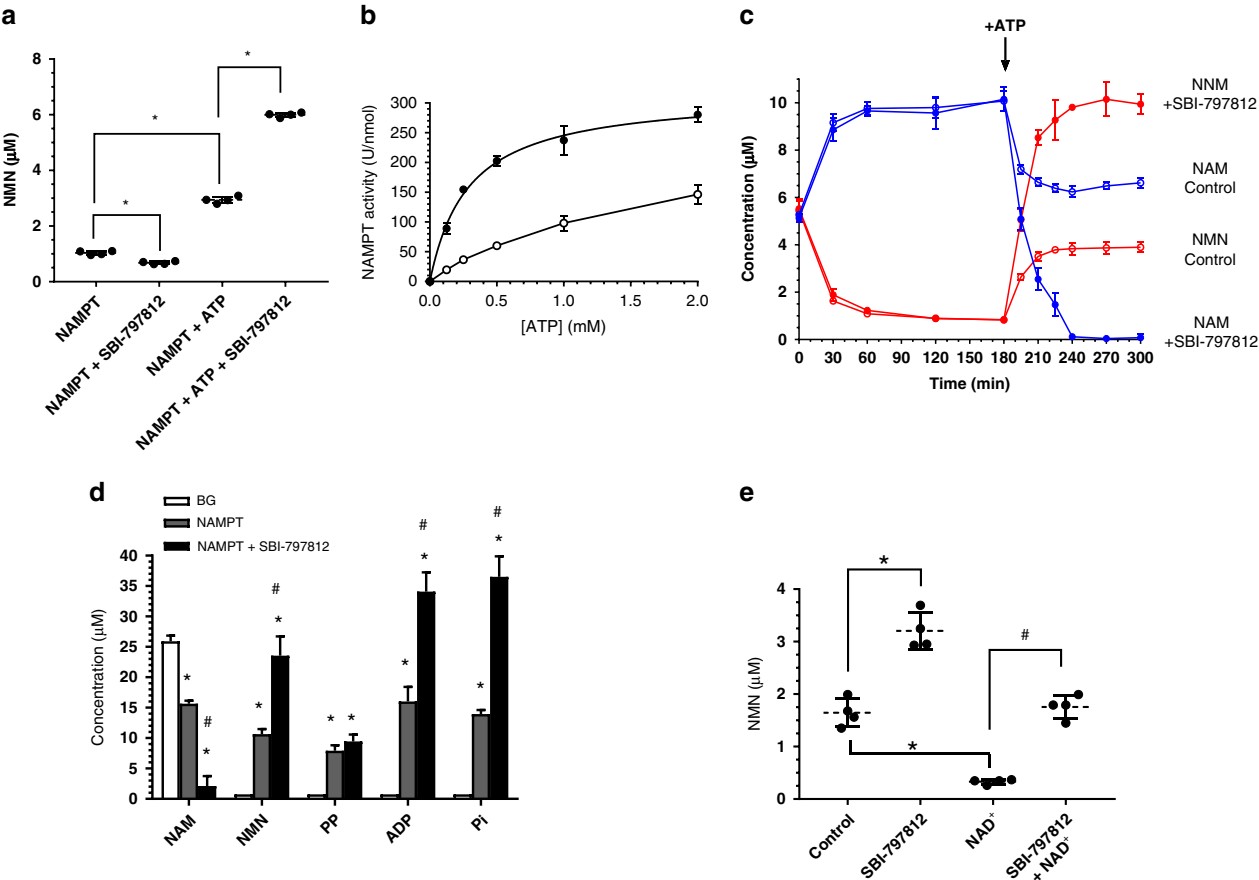

**Fig. 2** SBI-797812 mechanism of action. **a** NAMPT activation by SBI-797812 requires ATP. NAMPT (30 nM) incubated with NAM (10 μM) and PRPP (50 μM) and, where indicated, ATP (2 mM) and/or SBI-797812 (2 μM). Reactions performed at 37 °C for 1 or 4 h without or with ATP, respectively. NMN detected with the fluorescence assay. Data normalized to NMN produced in 1 h. Data expressed as means ± s.d.; $n = 4$. *$p < 0.001$ compared to NAMPT or (NAMPT + ATP). One-way ANOVA with Tukey's multiple comparisons test. **b** SBI-797812 increased NAMPT affinity for ATP. Without SBI-797812 (open circle), NAMPT (30 nM) incubated with NAM (10 μM), PRPP (50 μM), and varying ATP (0–2 mM). With 2 μM SBI-797812 (filled circle), reactions were identical except 15 nM NAMPT was used. Reactions performed for 1 h at 37 °C. NMN measured with the fluorescence assay. Data normalized for NMN production in the absence of ATP and expressed as U nmol$^{-1}$ NAMPT; means ± s.d. are shown; $n = 3$. Curve fitting using GraphPad Prism. **c** Impact of SBI-797812 on NAMPT equilibrium reaction. NAMPT (200 nM) mixed with 5 μM NAM, 100 μM PRPP, 5 μM NMN, 100 μM PP with (closed) and without (open) SBI-797812 (1 μM). Samples incubated at 37 °C and aliquots removed sequentially. At 3 h, 2 mM ATP was spiked. Incubation continued at 37 °C with aliquots sampled as indicated. NAM (blue) and NMN (red) assayed by LC-MS/MS. Data expressed as means ± s.d.; $n = 4$. **d** Comprehensive profiling of NAMPT reaction substrates/products. NAM (25 μM), PRPP (50 μM), and ATP (2 mM) incubated for 1 h at 37 °C in buffer without (BG) or with NAMPT (100 nM). Where indicated, SBI-797812 (5 μM) was included. Samples assayed for NAM, NMN and ADP by LC-MS/MS. Pi and PP determined by colorimetric assay. Data expressed as means ± s.d.; $n = 6$ except for BG (NAM, NMN, and ADP assays), $n = 5$. *$P < 0.0001$, vs. BG in each group; #$p < 0.0001$, NAMPT vs. (NAMPT + SBI-797812) in each group. 1-way ANOVA with Tukey's multiple comparisons test. **e** SBI-797812 blunted NAD$^+$-mediated feedback inhibition of NAMPT activity. NAMPT (30 nM) incubated with NAM (10 μM), PRPP (50 μM), ATP (2 mM) for 1 h at 37 °C. SBI-797812 (2 μM) or NAD$^+$ (250 μM) included as indicated. NMN measured by LC-MS/MS. Data expressed as means ± s.d.; $n = 4$. *$p < 0.0001$ vs. Control; #$p < 0.0001$ vs. NAD$^+$. 1-way ANOVA with Tukey's multiple comparisons test. For Fig. 2a–e, source data are provided as a Source Data file

solve the unrestricted transition state structure of the NAMPT-catalyzed reaction. Our in silico model (i.e., residues Asp313, Asp279, and pHis247, with PP, magnesium atoms and water molecules) generated a transition structure matching a late $S_N2$ attack by an activated PP on the pHis247 ($d_{N–P} = 2.38$ Å and $d_{P–O} = 1.96$ Å) to produce triphosphate (P3) (Fig. 3b).

An LC-MS/MS method to assay P3 was established to test this prediction. Incubation of NAMPT with ATP and PP produced P3 (Fig. 3c). P3 production was abolished in the presence of CHS-828. SBI-797812 stimulated P3 production by 3.8-fold (Fig. 3c). Notably, P3 was not produced when the NAMPT complete reaction (NAM, PRPP, NMN, PP) was performed. The explanation for this finding likely stems from the ability of NMN to blunt P3 production (Fig. 3c). While SBI-797812 reduced the levels of PP during the complete reaction (Fig. 2d), the actual fate of PP

coincident with concomitant NMN production requires further investigation.

Assaying the facultative ATPase activity of NAMPT (enzyme incubated with ATP only) yielded another clue regarding the impact of SBI-797812 on NAMPT catalytic activity. SBI-797812 increased the ATPase activity of NAMPT as judged by ADP production (Supplementary Fig. 6a). Interestingly, this was not matched by equimolar Pi production (Supplementary Fig. 6b) but instead gave rise to adenosine tetraphosphate (Ap4) (Supplementary Fig. 6c). Ap4 was recently shown to be a NAMPT product formed during ATP hydrolysis[22]. The ability of SBI-797812 to augment NAMPT-mediated production of either P3 (when incubated with ATP and PP) or Ap4 (when incubated with ATP) suggested that the reactivity of pHis247 in NAMPT was modulated by the NAMPT activator.

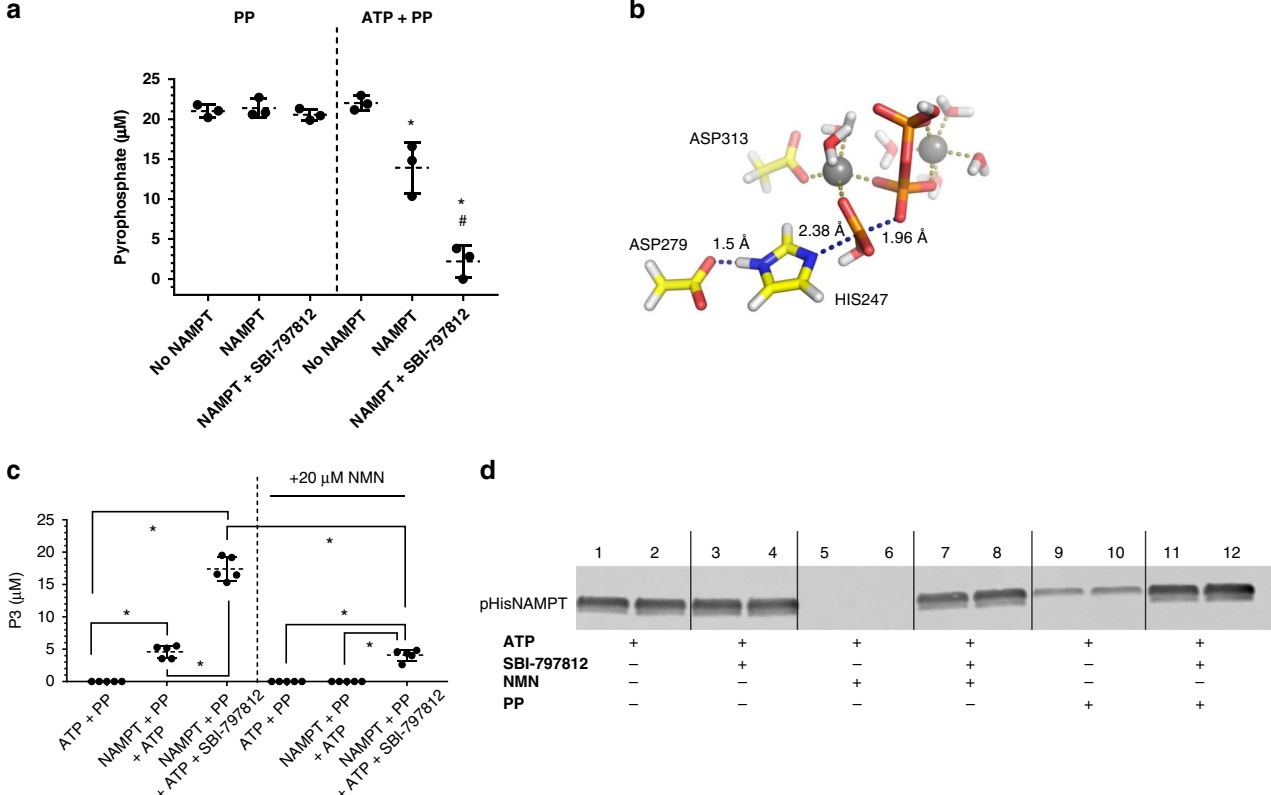

**Fig. 3** SBI-797812 impacts PP consumption and pHisNAMPT reactivity. **a** SBI-797812 stimulates NAMPT-mediated PP consumption. NAMPT (100 nM) was incubated with ATP (2 mM), PP (20 μM) or ATP + PP for 2 h at 37 °C in TMD buffer. Where indicated, SBI-797812 (5 μM) was also included. Samples assayed for PP using the colorimetric assay. Data expressed as means ± s.d.; $n = 3$. *$p < 0.001$ vs. No NAMPT (ATP + PP); #$p < 0.001$ vs. NAMPT (ATP + PP). One-way ANOVA with Tukey's multiple comparisons test. **b** Transition structure of the predicted reaction between pHisNAMPT and PP. The computational model used an original X-ray data set (PDB: 3DHF and 3DKL): pHis247, residues Asp313 and Asp279 are mimicked by acetate groups, two magnesium atoms (grey spheres) were incorporated to coordinate PP along with water molecules. The catalyzed reaction proceeds via late $S_N2$ attack of the pHis247 with $d_{N-P} = 2.38$ Å and $d_{P-O} = 1.96$ Å (blue dashed lines). **c** Production of P3 from ATP and PP by NAMPT. NAMPT incubated for 2 h with ATP (2 mM), PP (100 μM) in absence or presence of SBI-797812 (5 μM). Samples also run with 20 μM NMN where indicated. Samples analyzed by LC-MS/MS for P3. Data expressed as means ± s.d.; $n = 5$. *$p < 0.0001$. One-way ANOVA with Tukey's multiple comparisons test. **d** SBI-797812 stabilized pHisNAMPT in presence of NMN or PP. NAMPT (20 μg/ml) incubated with ATP (2 mM) for 10 min at 37 °C (all lanes). Also present were: 5 μM SBI-797812 (lanes 3–4), 10 μM NMN (lanes 5–6), NMN + SBI-797812 (lanes 7–8), 10 μM PP (lanes 9–10), PP + SBI-797812 (lanes 11–12). Samples were analyzed by western blotting using anti-1-pHisAb. pHisNAMPT was detected with a LICOR infrared imager. For Fig. 3a, c, d, source data are provided as a Source Data file

Our hypothesis that SBI-797812 altered the reactivity of pHis247 was supported by a western blotting approach that detected pHisNAMPT. pHisNAMPT was widely considered to be a highly-labile phosphoenzyme intermediate[11]. Autophosphory-lated NAMPT was previously detected but only as a weak signal by autoradiography after incubation of the enzyme with [γ-$^{32}$P]-ATP followed by SDS-PAGE[13]. We showed that NAMPT treated with ATP was visualized by western blotting using anti-1-pHis (δ$^1$-$N$) antibody (1pHisAb)[23] (Fig. 3d, Supplementary Fig. 7). NAMPT not exposed to ATP failed to react with 1pHisAb (Supplementary Fig. 7). Antibody raised against 3-pHis (ε$^3$-$N$) did not detect pHisNAMPT. The existence of 1-pHis (δ$^1$-$N$) but not 3-pHis (ε$^3$-$N$) in NAMPT agreed with the assignment from the crystal structure of NAMPT and beryllium fluoride (PDB: 3DHF), a putative pHisNAMPT mimic[10]. Consistent with the known chemical lability of 1-pHis(δ$^1$-$N$)[24], the pHisNAMPT band was abolished by heating the sample at 95 °C (Supplementary Fig. 7). While the pHisNAMPT band was suppressed by NMN or PP, NAM or PRPP had no effect (Fig. 3d, Supplementary Fig. 7). Incubation of NAMPT with ATP in the presence of SBI-797812 had little impact on pHisNAMPT accumulation (Fig. 3d). However, SBI-797812 stabilized pHisNAMPT in the presence of NMN or PP (Fig. 3d). We thus hypothesized that SBI-

797812 exerted a water shield effect whereby the NAMPT activator promoted the ability of nucleophiles other than water (e.g., PP) to abstract phosphate from the pHis247 residue.

Given the dramatic rightward shift of the NAMPT reaction equilibrium by SBI-797812, we explored if intracellular enzymes that might be "metabolic sinks" for NMN and PP (NMNAT1 and cytosolic inorganic pyrophosphatase PPA1, respectively) would compromise the SBI-797812 effect. NMNAT1 curtailed NMN accumulation in the assay because NMN was channeled towards NAD$^+$ synthesis (Supplementary Fig. 8). Importantly, SBI-797812 exerted comparable effects on (i) NMN production in the absence of NMNAT1 and (ii) combined NMN and NAD$^+$ production in the presence of NMNAT1. We next examined the impact of PPA1 in the absence and presence of SBI-797812 (Supplementary Fig. 9). The amount of PPA1 used in the assay was sufficient to completely degrade 20 μM PP in 10 min. PPA1 slightly "pulled" the NAMPT reaction towards NMN production (1.2-fold increase) in the absence of SBI-797812. Importantly, stimulation of NAMPT activity by SBI-797812 was not diminished in the presence of PPA1.

We also probed the effect of SBI-797812 on feedback inhibition of NAMPT activity by NAD$^{+}$[11]. Such suppression of NAMPT activity by NAD$^+$ which is formed in tandem with intracellular

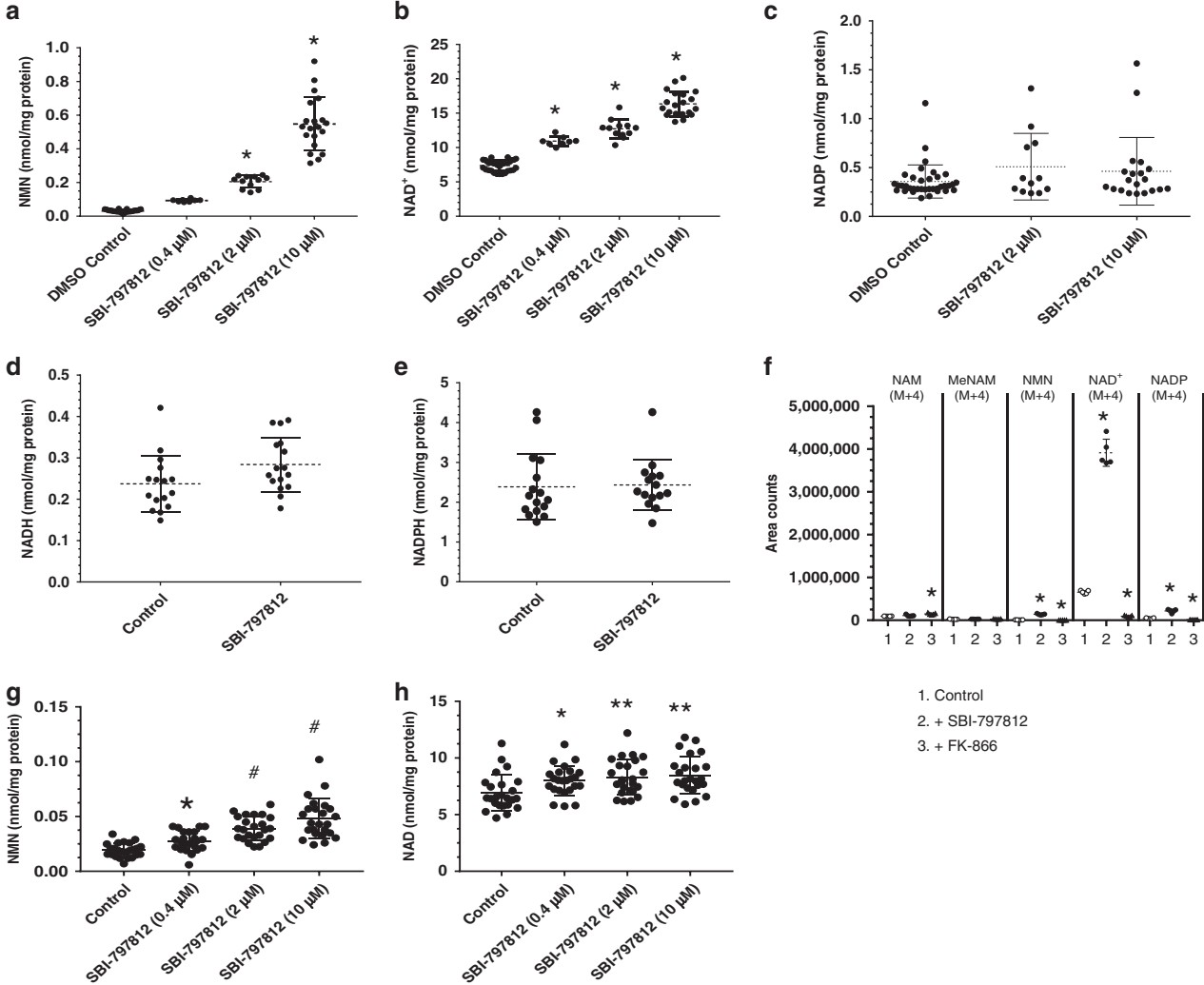

**Fig. 4** Effects of SBI-797812 on NAMPT activity in cultured cells. A549 cells treated with SBI-797812 for 4 h. NMN (**a**), NAD$^+$ (**b**), and NADP (**c**) were measured by LC-MS/MS. Data shows means ± s.d.; *$p < 0.0001$ vs. DMSO. NMN: $n = 35, 8, 12, 20$ for vehicle, 0.4, 2, 10 μM SBI-797812, respectively; NAD$^+$: $n = 32, 8, 12, 20$ for vehicle, 0.4, 2, 10 μM SBI-797812; NADP: $n = 36, 12, 20$ for vehicle, 2, 10 μM SBI-797812. One-way ANOVA with Tukey's multiple comparisons test. NADH (**d**) or NADPH (**e**) were not increased in A549 cells treated with SBI-797812 (10 μM) for 4 h. NADH and NADPH were assayed by LC-MS/MS. Data shows means ± s.d. $n = 16$ except for NADPH (SBI-797812), $n = 15$. **f** NAD$^+$ biosynthesis in A549 cells treated for 4 h with vehicle, SBI-797812 (10 μM) or FK-866 (1 μM), and $^{13}C/^{15}N$-NAM (20 μM). NAM(M + 4), MeNAM(M + 4), NMN(M + 4), NAD$^+$(M + 4), and NADP(M + 4) were assayed by LC-MS/MS. Data shows means ± s.d.; $n = 5$. *$p < 0.001$ vs. Control. One-way ANOVA with Dunnett's multiple comparisons test. **g**, **h** SBI-797812 increased NMN and NAD$^+$ in human primary myotubes. Cells treated with SBI-797812 (0.4, 2, 10 μM) for 4 h. NMN and NAD$^+$ quantified by LC-MS/MS and normalized to cell protein. Data shows means ± s.d.; $n = 24$. *$p < 0.05$, **$p < 0.01$, #$p = 0.0001$ vs. Control. One-way ANOVA with Dunnett's multiple comparisons test. **e**. **f** For Fig. 4a–h, source data are provided as a Source Data file

NMNAT activity would stymie NAD$^+$ booster strategies. Interestingly, SBI-797812 relieved NAMPT inhibition mediated by NAD$^+$ (Fig. 2e). This unanticipated impact of the small molecule NAMPT activator should further promote intracellular production of NMN and NAD$^+$.

**Cellular effects of the NAMPT activator**. We used A549 human lung carcinoma cells for routine testing of NAMPT activators. Exposure of A549 cells to SBI-797812 for 4 h produced dose-dependent elevations of intracellular NMN (Fig. 4a) and NAD$^+$ (Fig. 4b). The NMN level in A549 cells was $30 ± 7$ pmol mg$^{-1}$ protein. The fold elevations of NMN were 2.7, 6.1, and 16.7 in the presence of 0.4, 2, and 10 μM SBI-797812, respectively. The level of NAD$^+$ in A549 cells was $7.4 ± 0.8$ nmol mg$^{-1}$ protein. The fold elevations of NAD$^+$ were 1.5, 1.7, and 2.2 in the presence of 0.4,

2, and 10 μM SBI-797812, respectively. SBI-7979812 had no significant impact on the levels of NADP (Fig. 4c), NADH (Fig. 4d) or NADPH (Fig. 4e) in A549 cells. The apparent decreased potency of SBI-797812 in the cellular assays likely reflects binding by intracellular proteins and serum-containing cell culture media.

To further explore the impact of SBI-797812 on NAD$^+$ synthesis, A549 cells were treated with $^{13}C/^{15}N$-labelled NAM [NAM(M + 4)] and intracellular levels of NAM(M + 4), 1-methylnicotinamide (1-MeNAM)(M + 4), NMN(M + 4), NAD$^+$(M + 4) and NADP (M + 4) were measured (Fig. 4f). The study was performed in the absence or presence of SBI-797812 or FK-866 for 4 h. Cellular NAM(M + 4) did not accumulate, except for a small but significant increase when cells were also treated with FK-866. No differences in the levels of 1-MeNAM(M + 4) were observed. The meager appearance of intracellular NADP(M + 4) revealed that conversion of NAD$^+$ to NADP by NAD$^+$ kinase was not a dominant pathway

in A549 cells. Intracellular NMN(M + 4) was slightly increased in the presence of SBI-797812 and decreased in the presence of FK-866. By far, the $^{13}C/^{15}N$-labelled NAM-containing species that displayed the largest intracellular accumulation was $NAD^+$(M + 4). SBI-797812 dramatically increased the level of $NAD^+$(M + 4) by 5-fold as compared to control whereas $NAD^+$(M + 4) production was abolished by FK-866. Importantly, this experiment unveiled a more robust effect of SBI-797812 on NAMPT activity than inferred by assaying the total intracellular $NAD^+$ pool. This finding undoubtedly reflects cellular homeostatic mechanisms that exert a ceiling effect on the intracellular $NAD^+$ levels.

We also examined the impact of SBI-797812 on intracellular levels of NMN (Fig. 4g) and $NAD^+$ (Fig. 4h) in human primary myotubes. Treatment of these cells for 4 h with SBI-797812 elicited dose-dependent increases in the intracellular levels of NMN and $NAD^+$. SBI-797812 at 10 μM elicited 2.5- and 1.25-fold increases of intracellular NMN and $NAD^+$, respectively. Exposure of mouse primary myotubes to SBI-797812 (10 μM) for 4 h also elicited significant increases of NMN and $NAD^+$ (Supplementary Fig. 10).

Recent assessment of $NAD^+$ biosynthetic flux in cultured cells concluded that PARP1/2 and SIRT1/2 are major $NAD^+$ consumers and their cellular activities are governed by the intracellular $NAD^+$ concentration[7]. Hence, elevated intracellular

$NAD^+$ mediated by SBI-797812 might elicit concomitant activation of sirtuins and PARPs. To probe a possible impact of SBI-797812 on sirtuin activity, we examined acetylation of histone H4, a known SIRT1 target[25]. Addition of SBI-797812 to A549 cells for 4 h decreased the H4-AcK16/H4 ratio (Fig. 5a), an effect that is consistent with SIRT1 activation. We next examined a possible impact of SBI-797812 on PARP-1 activity. PARP-1 activation elicits conspicuous auto-PARylation as detected by western blotting[26]. Addition of SBI-797812 to A549 cells for 4 h did not increase auto-PARylated PARP-1 whereas treating A549 cells with $H_2O_2$, a PARP-1 activation trigger, increased auto-PARylated PARP-1 (Fig. 5b). SBI-797812 did not alter the auto-PARylated PARP-1 level in the presence of $H_2O_2$. However, when cell lysates were prepared in the absence of a PARP-1 inhibitor (used to block post-cell lysate PARylation artifacts), lysates from cells treated with SBI-797812 displayed markedly elevated auto-PARylated PARP-1 (Fig. 5c). This result established that increased intracellular $NAD^+$ levels in A549 cells treated with SBI-797812 promoted PARP-1 activity in the cellular lysate. The lack of an SBI-797812 effect on PARP-1 activation in intact cells might reflect subcellular compartmentalization constraints or the dissimilar substrate ($NAD^+$) concentrations in intact cells versus cellular lysates (this latter explanation relates to the $K_m$ value of PARP-1 for $NAD^+$)[27].

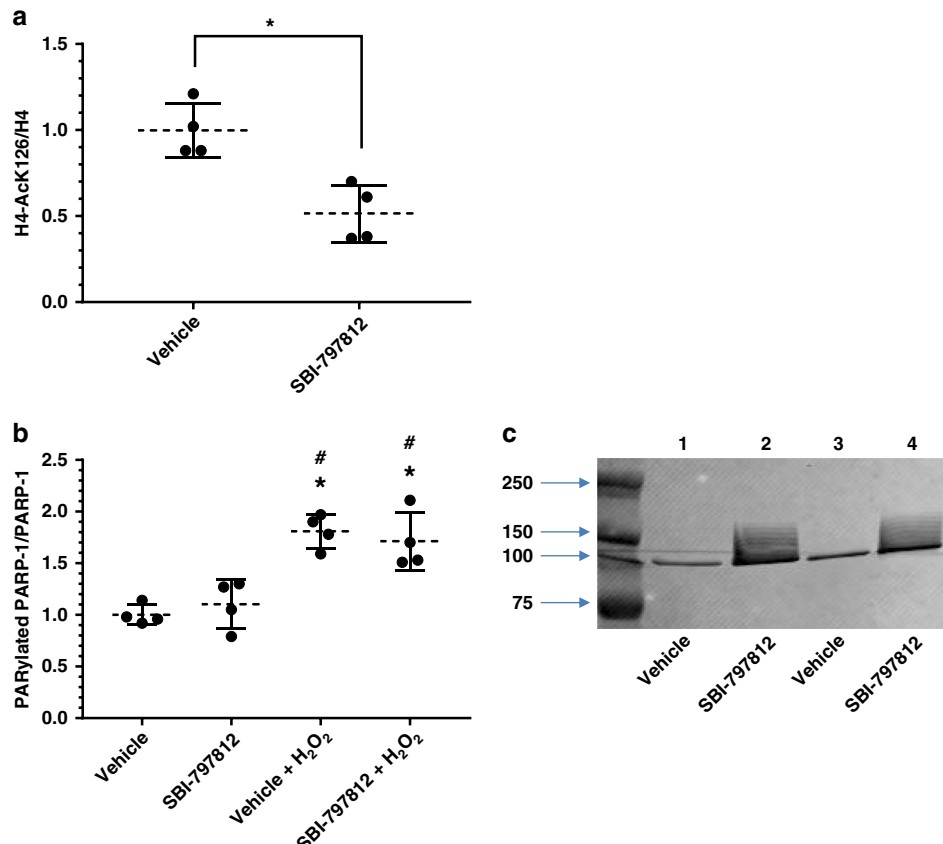

**Fig. 5** Exploring the impact of SBI-797812 on sirtuin and PARP-1 activities. **a** SBI-797812 decreases histone H4 acetylation. A549 cells treated with vehicle or SBI-797812 (10 μM) for 4 h. Histone H4 and H4-AcK16 levels determined by western blotting. Normalized signal ratios are shown. *$p = 0.005$ vs. control; $n = 4$. Two-tailed unpaired $t$ test used. **b, c** Impact of SBI-797812 on PARP-1 activation. A549 cells treated with vehicle or SBI-797812 (10 μM) for 4 h. Where indicated, A549 cells were also treated with $H_2O_2$ (0.5 mM) for 30 min before harvest. Cell lysates were prepared with (**b**) or without (**c**) olaparib and analyzed by western blotting. Cell lysates probed with PAR and PARP-1 antibodies (**b**). Signal intensities determined with Licor Imaging system. PAR:PARP-1 signal ratios normalized to vehicle, no $H_2O_2$ (assigned value = 1). Data shows means ± s.d., $n = 4$. *$p < 0.002$ vs. vehicle; #$p < 0.01$ vs. SBI-797812. One-way ANOVA with Tukey's multiple comparisons test. Panel c shows western blotting of A549 cells treated with vehicle or SBI-797812 (10 μM) for 4 h; cell lysates prepared without olaparib and probed with anti-PAR antibody. For Fig. 5a–c, source data are provided as a Source Data file

**In vivo effects of NAMPT activators in mice**. SBI-797812 was administered to mice (10 mg compound kg$^{-1}$ body weight) by oral or intraperitoneal (i.p.) dosing, blood was drawn at increasing times and plasma levels of SBI-797812 were measured by LC-MS/MS. Plasma concentrations of SBI-797812 after oral administration were low (Supplementary Fig. 11a). Higher plasma levels of SBI-797812 were seen after i.p. dosing ($C_{max}$ value: 3297 ng ml$^{-1}$, 8.2 µM) (Supplementary Fig. 11b). The transient plasma exposure of SBI-797812 was a key consideration when evaluating the pharmacodynamic effects of SBI-797812 in mice.

The potency of SBI-797812 against mouse NAMPT was also investigated. The specific activity (U nmol$^{-1}$ NAMPT) of mouse NAMPT using the in vitro NMN production assay was approximately eight-fold higher than human NAMPT. Moreover, the apparent affinity ($EC_{50}$) of SBI-797812 for mouse NAMPT was approximately 8-fold less than for human NAMPT, whereas maximal fold activation by SBI-797812 was comparable between the mouse and human NAMPT. These differences between human and mouse NAMPT are another important factor when evaluating the in vivo efficacy of SBI-797812 in murine preclinical models.

For the tissue biomarker study, mice were dosed with SBI-797812 (20 mg kg$^{-1}$ i.p.) and liver, heart, gastrocnemius muscle and quadriceps muscle were harvested after 4 h. Despite the transient plasma exposure of SBI-797812 and lower apparent affinity of SBI-797812 for mouse NAMPT, a significant 1.3-fold increase of NAD$^+$ was detected in liver (Fig. 6a). There was also a trend towards increased NAD$^+$ levels in cardiac tissue (Fig. 6b). Skeletal muscle, either gastrocnemius (Fig. 6c) or quadriceps (Fig. 6d), did not exhibit increased NAD$^+$ levels after dosing with SBI-797812. The mean tissue levels of SBI-797812 (2 h post-dose) as measured by LC-MS/MS were 0.311, 0.144, 0.078, and 0.078 µg/mg dry powder in liver, heart, gastrocnemius and quadriceps, respectively (Supplementary Fig. 12). Hence, liver which

displayed a statistically significant increase of NAD$^+$ after SBI-797812 dosing exhibited the highest level of the compound.

## Discussion

Our search for small molecules that stimulated NAMPT-mediated NMN formation yielded SBI-797812 which activated purified NAMPT, raised NMN and NAD$^+$ levels in cultured cells, and boosted hepatic NAD$^+$ in mice. The structural similarity between SBI-797812 and active-site targeted NAMPT inhibitors was striking. Remarkably, moving the pyridyl nitrogen from the 4-position in SBI-797812 to the 2 or 3-positions changed the compound from a NAMPT activator to an inhibitor. Binding of SBI-797812 to the NAMPT active site was supported by two independent experimental approaches. Firstly, the NAMPT (G217R) mutant was refractory to activation by SBI-797812, just as it was insensitive to inhibition by CHS-828[18] and FK-866. Secondly, FK-866 and CHS-828 blocked binding of SBI-797812 to NAMPT as determined with a "spin column" method that separated free ligand from NAMPT-bound ligand. The fact that SBI-797812, an active-site targeted ligand, was a NAMPT activator rather than an inhibitor posed a vexing yet fascinating conundrum.

SBI-797812 had a dramatic effect on the equilibrium position of the reversible NAMPT reaction eliciting a marked elevation of NMN and concomitant depletion of NAM. Several clues to the SBI-797812 MOA have been gleaned from our investigation (Fig. 7). First, ATP was obligatory for the activating effect of SBI-797812. SBI-797812 increased the apparent affinity of NAMPT for ATP. Second, SBI-797812 stabilized pHisNAMPT in the presence of NAMPT products (NMN or PP) as shown by western blotting using an 1pHisAb. Stabilization of pHisNAMPT should manifest itself as increased affinity for NAMPT substrates[11]. The water shield effect of SBI-797812 on pHis247 was likely

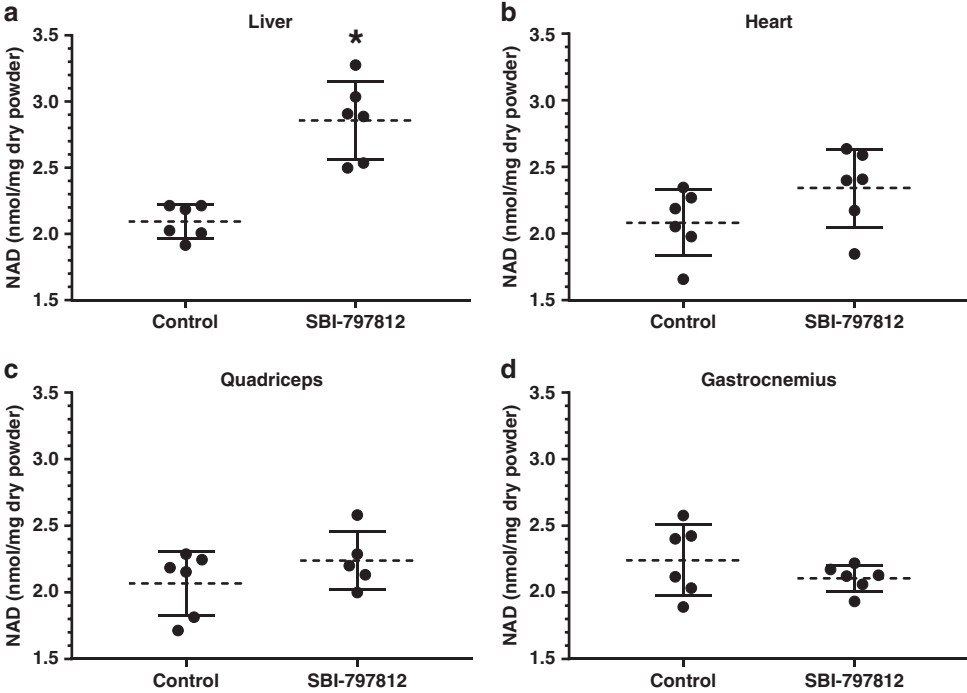

**Fig. 6** Effects of SBI-797812 on tissue NMN and NAD$^+$ in mice. SBI-797812 (20 mg kg$^{-1}$) or vehicle were dosed to mice by i.p. administration. Liver (**a**), heart (**b**), gastrocnemius (**c**), and quadriceps (**d**) were harvested at 4 h post-dosing. Tissues were assayed for NAD$^+$ (which was normalized to mg of dry tissue) by LC-MS/MS. Data shows means ± s.d.; $n = 6$ except for quadriceps (SBI-797812), which was $n = 5$. *$p = 0.0002$ vs. Control. Two-tailed unpaired $t$ tests were used. Source data are provided as a Source Data file

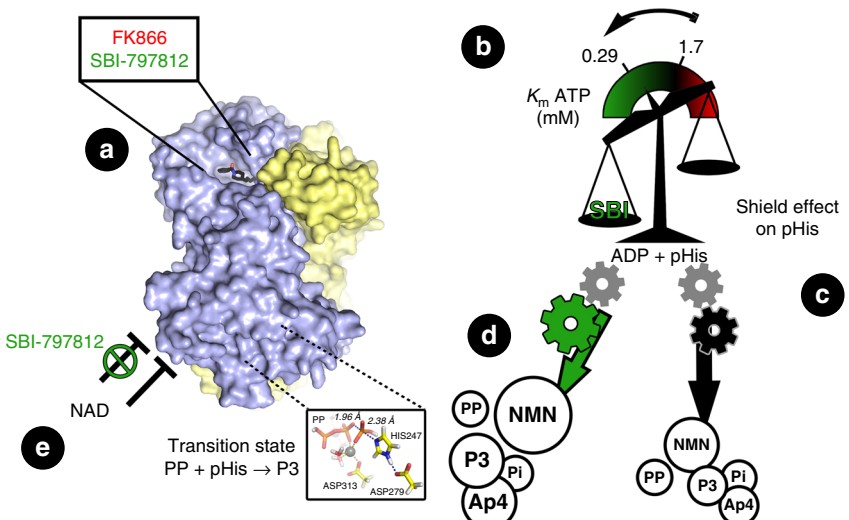

**Fig. 7** Proposed SBI-797812 mechanism of action. SBI-797812 (NAMPT activator) and FK-866 (NAMPT inhibitor) have overlapping binding sites at the NAMPT active site (**a**). Binding of SBI-797812 dramatically shifted the reaction equilibrium of the reversible NAMPT reaction to promote NMN production and NAM consumption. ATP was obligatory for NAMPT activation by SBI-797812. SBI-797812 increased the affinity of NAMPT for ATP (**b**). SBI-797812 exerted a Water Shield Effect on the phosphorylated i.e., activated) form of NAMPT (pHisNAMPT) (**c**). The pHis247 adduct was more resilient to hydrolysis upon SBI-797812 binding (green gear) such that the PRPP/NAM, ATP and PP substrates can better react with pHisNAMPT. This leads to increased levels of NMN (in the "complete" reaction), P3 (in the presence of ATP + PP) or Ap4 (in the presence of ATP) (**d**). The transition state for NAMPT-catalyzed P3 formation is depicted in the insert. Finally, SBI-797812 binding relieved $NAD^+$ end product feedback inhibition (in the presence of NMNAT1-3) (**e**). Overall, SBI-797812 transformed NAMPT into a "super catalyst" making it possible for substrates to be captured and processed more efficiently by the enzyme

responsible for the remarkable ability of the NAMPT activator to promote formation of P3 and Ap4 from the nucleophiles PP and ATP, respectively. Our observations that NMN both abolished P3 production and blocked the appearance of pHisNAMPT (when NAMPT was incubated with ATP) are probably inextricably linked. Third, NAMPT consumed the PP by-product in a reaction that was stimulated by SBI-797812. The fate of PP in the presence of NMN remains to be determined but PP consumption would certainly contribute to the "rightward shift" of the NAMPT reaction.

We established that the ability of SBI-797812 to activate purified NAMPT was recapitulated in cultured cells. While the fold-increase of NMN in SBI-797812-treated A549 cells (17.4-fold) was much larger than the increase of $NAD^+$ (2.2-fold), the total $NAD^+$ rise greatly exceeded the total NMN rise by approximately 17.5-fold. Hence, NMNAT did not emerge as the rate-limiting step when NAMPT was activated with SBI-787812.

In an intracellular milieu where NAMPT works in tandem with NMNAT1-3 to produce $NAD^+$, the interaction of SBI-797812 with NAMPT mediated two other effects that should also promote $NAD^+$ accumulation. First, SBI-797812 blunted feedback inhibition of NAMPT activity by $NAD^+$, thus opposing product inhibition of $NAD^+$ synthesis. Suppression of feedback inhibition by $NAD^+$ with a small molecule NAMPT activator might be a key advantage for raising intracellular $NAD^+$ as compared to nicotinamide riboside (NR), an NMN precursor[5,28]. Second, colocalization of NAMPT and NMNAT in various subcellular compartments[29] suggests that PP produced by the NMNAT reaction might be also consumed by NAMPT in a process stimulated by SBI-797812. As the equilibria of the NMNAT reactions favor $NAD^+$ catabolism[30], such PP depletion by NAMPT would act to further displace the NAM salvage pathway equilibrium towards $NAD^+$-end product formation.

The vast arsenal of intracellular enzymes that consume $NAD^+$ constitutes a powerful homeostatic mechanism that exerts a ceiling effect on cellular $NAD^+$ levels. The activities of PARPs

and sirtuins in T47D cultured cells were shown to be regulated by the intracellular $NAD^+$ concentration (i.e., $NAD^+$ elevation is opposed by increased $NAD^+$ consumption)[7]. To better recognize the ability of SBI-797812 to activate NAMPT in cells, we co-treated cells with $^{13}C/^{15}N$-NAM and the ensuing NAM-containing metabolites were monitored for the presence of this stable isotope tracer. The dominant fate of the NAM tracer in A549 cells was the $NAD^+$ pool. Importantly, SBI-797812 dramatically elevated the intracellular $NAD^+$(M + 4) level. The NAM tracer was not enriched in the 1-MeNAM pool which is formed by nicotinamide N-methyltransferase (NNMT), a key regulatory enzyme in numerous cell types[31]. The rise of NMN(M + 4) was very slight compared to $NAD^+$(M + 4) which suggested that NMNAT activity was not rate determining even when NAMPT activity was increased by SBI-797812. There was very modest appearance of NADP(M + 4). This result was concordant with the recent $NAD^+$ flux study showing that NADP was slowly labeled in T47D cells[7].

Achieving the desired pharmacological effect when dosing mice with a NAMPT activator is contingent on several prerequisites. Firstly, compound dosing must yield adequate plasma exposure. The pharmacokinetic profile of SBI-797812 was mediocre but appreciable plasma exposure was seen up to 4 h after i.p. dosing. Secondly, the NAMPT activator must cross-react with murine NAMPT. This was the case even though SBI-797812 was a weaker activator of mouse NAMPT as compared to human NAMPT. Thirdly, $NAD^+$ biosynthetic flux in target tissues must be rapid relative to the study duration. This prerequisite appeared to be met given the ability of SBI-797812 to elevate NMN and $NAD^+$ in A549 cells at 4 h post-dosing. However, the smaller increases of NMN and $NAD^+$ in primary mouse myotubes portended that responsiveness to SBI-797812 was not uniform across different cell types. This latter concern was heightened by recent data showing that mammalian $NAD^+$ metabolism involves extensive tissue-specific pathway regulation, which is not replicated in standard cell lines[7].

Despite these shortcomings of SBI-797812, administration of our prototypical NAMPT activator to mice increased hepatic NAD$^+$ levels. Future studies with superior compounds will allow us to decipher if the relatively modest in vivo NAD$^+$ elevating effects of SBI-797812 can be improved with analogs that exhibit more favorable pharmacokinetics and/or potency versus murine NAMPT. In addition, changes in the study design such as longer duration of dosing and interrogation of other tissues may also unveil more robust effects of a NAMPT activator. It is also possible that the NAD$^+$ levels are not an ideal biomarker for NAMPT activation due to cellular mechanisms that divert the newly- generated NAD$^+$ to other metabolic by-products as strongly suggested by our cell study using the stable-isotope NAM tracer.

Comparing the therapeutic utility of small molecule NAMPT activators to NAD$^+$ boosters such as NR or NMN will be essential. The fact that SBI-797812 acts catalytically to promote NAD$^+$ synthesis along with its ability to suppress feedback inhibition of NAMPT activity by NAD$^+$ (Fig. 2e) are two discriminating attributes that are likely to be advantageous. In any event, a small molecule NAMPT activator exemplified by SBI-797812 represents a pioneering pharmacological approach to raise intracellular NAD$^+$ and realize diverse and potentially impactful therapeutic benefits.

## Methods

**Materials and reagents**. Commercially-available materials and reagents are listed in Supplementary Table 1.

**Custom produced recombinant proteins**. Human NAMPT (N-terminal His-tagged) was expressed in *E.coli* BL21 (DE3) pLysS containing expression plasmid (pBAD-DEST 49) with DNA encoding human NAMPT[11]. Cells were harvested and lysed with a pneumatic high shear fluid homogenizer (Microfluidizer LM10, Microfluidics) in lysis buffer (PBS containing 10 mM imidazole, pH 7.4). The lysed sample was centrifuged (30 min at 13,000 rpm) and the supernatant was applied to Complete His-tag purification resin (Roche Diagnostics). The column was washed with PBS buffer (pH 7.4) containing 20 mM imidazole and His-tagged NAMPT was eluted with 0.5 M imidazole in PBS. The penultimate purification step used a HiTrap Q column (GE Healthcare) equilibrated with 20 mM Tris buffer (pH 8.0). NAMPT was eluted with a 0–500 mM NaCl gradient. NAMPT-containing fractions were concentrated and treated with thrombin (5 IU thrombin for each 1 mg NAMPT at room temperature for 2 h) to remove the N-terminal His tag extension. Final NAMPT purification involved re-chromatography with the HiLoad 16/60 column (100 mM HEPES pH 7.5, 100 mM NaCl and 10 mM 2-mercaptoethanol). Purified human NAMPT was stored at −80 °C. Protein concentration was determined by amino acid composition analysis performed by the Proteomics and Mass Spectrometry Facility at the Donald Danforth Plant Science Center (St. Louis, MO). Mouse NAMPT was expressed in *E.coli* and purified similarly. The human NAMPT(G217R) mutant was cloned from the NAMPT WT cDNA and expressed/purified essentially as above.

Human *NMNAT1* was synthesized in *E.coli* BL21(DE3) containing expression plasmid pET20b (+) with cDNA encoding NMNAT1. The custom-synthesized oligonucleotide primers used for PCR-mediated amplification of the NMNAT1 cDNA (35 cycles; 56 °C annealing temperature) are shown in Supplementary Table 2. Cells were lysed and His-tagged NMNAT1 was isolated using the Complete His-tag purification resin (Roche Diagnostics) as described above. The NMNAT1-containing fractions were pooled, dialyzed against 20 mM Tris-HCl, 500 mM NaCl, 3 mM DTT, 10% glycerol (pH 7.4) and stored at −80 °C.

**Protein thermal shift assay**. The Conrad Prebys Center for Chemical Genomics (CPCCG) at SBP-La Jolla screened 57,004 compounds (ChemBridge Premium Set; 10 mM compound stocks in DMSO) to identify chemical structures that caused a thermal shift of purified human NAMPT. The protein thermal shift (PTS) assay was performed in 384 well plates (10 µl well volume). The PTS assay used 2 µM NAMPT, 5x Sypro Orange dye (Molecular Probes), 2.5 mM ATP in 50 mM HEPES, pH 7.5 buffer containing 50 mM NaCl, 5 mM MgCl$_2$ and 1 mM TCEP. Compounds (25 µM) were added using an Echo acoustic dispenser and incubated for 15 min prior to the assay. The final DSMSO concentration was 0.25%. The assay was performed with a ViiA7 real time PCR system (Thermo Fisher Scientific) at 0.15° C sec$^{-1}$ ramping speed. The temperature ranged from 25 to 95 °C. The primary screen hit criterion was Z-score ≥ 5 or $\Delta T_m$ $D \geq 1$ °C. Hit confirmation was performed with fresh powders and used the following PTS criterion: Z-score ≥ 7 or delta T$_m$ $D \geq 1$ °C (triplicate wells at a compound concentration of 25 µM). Further details of the PTS assay are presented in Supplementary Table 3.

**Detection of NAMPT activators in PTS hit set**. PTS hits were tested for their abilities to stimulate NAMPT activity with a 2-step sequential assay. Firstly, NAMPT (16 nM), NAM (5 µM), PRPP (6.25 µM), ATP (0.12 mM), yeast inorganic pyrophosphatase (0.04 U ml$^{-1}$) and NMNAT3 (5 µg ml$^{-1}$) with or without PTS hits (25 µM final concentration; performed in triplicate) were incubated for 2 h at room temperature in 50 mM HEPES, pH 7.5, 50 mM NaCl, 5 mM MgCl$_2$, 1 mM TCEP, 0.005% Tween 20 (1536 well format). Next, NAD$^+$ was quantified with NAD-Glo assay kits (Promega Corp). The NAD-Glo master reagent was added and luminescence was detected after 30 min at room temperature.

**NAMPT enzymatic reactions**. The NMN production assay involved incubation of NAMPT at 37ºC with NAM (25 µM), PRPP (50 µM) and ATP (2 mM) at 37 °C in TMD buffer (50 mM Tris-HCl, 10 mM MgCl$_2$, 2 mM DTT, pH 7.5). Where indicated (see Fig. 2c), NMN and PP were also included. Also, where specified (see Fig. 2b), the ATP concentration was varied from 0 to 2 mM. The values for $V_{max}$ and $K_m$ (ATP hydrolysis) were deduced using the on-line Michaelis-Menten kinetics tool at http://www.graphpad.com/quickcalcs/ttest1/?Format=SEM. The reactions were performed in the absence or presence of NAMPT activators, inhibitors or other agents as specified. The final DMSO concentration was 1%. NAM, NMN and ADP were assayed by LC-MS/MS (see Supplementary Information) after quenching samples with equal volume of 1 M perchloric acid (PCA). NMN was also assayed using a chemical method which converts NMN into a fluorescent derivative[19]. For the latter assay, an aliquot (37.5 µl) of the NMN-containing sample was sequentially mixed with 15 µl of 20% acetophenone (in DMSO) and 15 µl of 2 M KOH. The mixture was placed on ice for 10 min. Next, 67.5 µl of 100% formic acid was added to each sample, vortexed, and then incubated at 37 °C for 20 min. Samples (100 µl) were transferred to a 96-well opaque bottom plate and fluorescence (Ex/Em = 382/445 nm) was measured using a SpectraMax M5 plate reader (Molecular Devices). Pi was assayed using the PiColorLock Gold Phosphate detection system (Innova Biosciences). PP concentration was calculated from the difference in the amount of Pi without and with treatment with yeast inorganic pyrophosphatase (40 ng ml$^{-1}$ for 10 min at 37 °C).

The PP consumption assay involved incubation of NAMPT at 37 °C with PP, ATP and other agents as indicated. PP was measured using the PiColorLock Gold Phosphate Detection System as described above. P3 was measured by LC-MS/MS (see Supplementary Information).

The ATPase assay involved incubation of NAMPT at 37ºC with 2 mM ATP in TMD buffer and other agents as indicated. ADP was measured by LC-MS/MS (see Supplementary Information). Pi was measured using the PiColorLock Gold Phosphate Detection system.

**SBI-797812 binding to NAMPT**. Human NAMPT (2 µM) and SBI-797812 (5 µM) were combined in T8MD-Tw buffer (50 mM Tris pH 8.0, 10 mM MgCl$_2$, 2 mM DTT, 0.01% Tween 80). Where indicated, NAMPT was first treated with FK-866 (5 µM) or CHS-828 (5 µM). Samples were incubated at 37 °C for 10 min and then placed on ice. Samples (60 µl) and T8MD-Tw buffer (15 µl) were sequentially added to Zeba Spin Desalting Columns (7 K MWCO; 89883, Thermo Fisher Scientific) equilibrated with T8MD-Tw buffer. Columns were centrifuged at $1500 \times g$ at 4 °C for 2 min. Eluents were collected and frozen at −80 °C. SBI-797812 was assayed as follows. Samples (25 µl) were extracted with 100 µl acetonitrile (ACN) containing 1 µg ml$^{-1}$ indomethacin as the internal standard (IS). Samples were vortexed 5 min, centrifuged 3700 x rpm at 4 °C for 10 min, and 100 µl aliquots of each supernatant were transferred to a 96-well plate. Ten microliter aliquots of the extracts were injected onto a Thermo HPLC system equipped with PAL CTC plate sampler (96-well plate), Dionex Ultimate 3000 binary pump (flow rate at 0.25 ml min$^{-1}$), Dionex Ultimate 3000 thermostatted column compartment (temperature held at 40 °C), Thermo Endura Mass Spectrometer (ESI source), and Thermo Scientific Accucore C18 (2.6 µm, 2.1 × 50 mm, 100 Å) column. The HPLC solvents were ACN / 0.1% formic acid (A) and 0.1% formic acid (B). The column gradient was: 5 to 95% A from 0 to 5.0 min, 95% A until 5.5 min and step reduction to 5% A for 1 min for column reequilibration. SBI-797812 peak areas were measured, and analyte amounts were calculated from calibration curves after adjusting for IS concentrations. SBI-797812 calibration curves were constructed with 8 concentrations (1, 5, 10, 50, 100, 500, 1000, and 5000 nM) by spiking 10 µl of 50x concentration DMSO stocks into 490 µl buffer, extracting 25 µl of the resulting sample and analyzing as described above.

**Western blotting for pHisNAMPT**. pHisNAMPT was produced by treating NAMPT (10 µg ml$^{-1}$) with 2 mM ATP in TMD buffer. Where indicated, other agents (SBI-797812, NAM, PRPP, NMN, PP) were also included. Samples were incubated at 37 °C for 10 min and combined with SDS-containing sample preparation buffer. Samples were kept on ice and not heated prior to loading on the gel. Two hundred ng of protein was run on a 4–20% Criterion TGX gel (Bio-Rad) at 150 V for 1.5 h and transferred to a polyvinylidene fluoride (PVDF) membrane (Roche) at 100 V for 30 min. Both the gel running and PVDF transfer steps were performed at 4 °C. PVDF membranes were treated with Blocking Buffer (LI-COR Bioscience) for 1 h at 4 °C, and then exposed to rabbit monoclonal anti-N1-phosphohistidine (1-pHis) (Millipore Sigma, cat no. MABS1330, clone SC1-1) 1:500-dilution for 16 h at 4 °C. PVDF membranes were washed and treated with

IRDye 800CW goat anti-rabbit IgG (LI-COR Biosciences) for 1 h at 4 °C. PVDF membranes were washed and bands were visualized with an Odyssey Digital Infrared Imaging System (LI-COR Biosciences).

**Cellular studies.** Human A549 lung carcinoma cells (American Type Culture Collection; Manassas, VA, USA) were grown in DMEM, 4.5 g l$^{-1}$ D-(+)-glucose, 10% fetal bovine serum, penicillin/streptomycin mixture; 10 cm dishes) and treated with DMSO (vehicle control) or SBI-797812 for 4 h. Cells were washed with cold PBS and immersed in liquid nitrogen. After decanting the liquid nitrogen, the cells were scraped, collected and stored at −80 °C. In a related protocol, A549 cells were exposed to media also containing 20 μM NAM ($^{13}C_3$, $^{15}N$) (Cambridge Isotope Laboratories) along with vehicle or SBI-797812 for 4 h. Cells were harvested and processed as described above. Intracellular NMN, NAD$^+$, NADP, NADH and NADPH were quantitated by LC-MS/MS (see Supplementary Information). LC-MS/MS was also used to measure the levels of NAM-containing metabolites that possessed the NAM(M + 4) moiety. Different cell extraction protocols were used for oxidized and reduced pyridine nucleotides. For oxidized pyridine nucleotides, 10 μl of the thawed cells was quickly removed for the protein assay, and 200 μl 1 M PCA was added to the remainder. Samples were vortexed, centrifuged and volumes were recorded. Samples were transferred to microfuge tubes and volumes were normalized to 400 μl with distilled water. A 100 μl aliquot was removed and used for targeted metabolite profiling. Total cellular protein was measured using the Pierce BCA Protein assay kit (ThermoFisher Scientific) with BSA as the standard. Reduced pyridine nucleotides were assayed similarly except that that 500 μl 50:50 0.1 M NaOH/MeOH was used to lyse the cells.

**Western blotting for sirtuin and PARP-1 activities.** A549 cells (10 cm dishes) were treated with vehicle or SBI-797812 (10 μM) for 4 h as described above. Where indicated, 0.5 mM H$_2$O$_2$ was also added to the cells for 30 min (from 3.5 to 4 h). Cell lysis buffer contained RIPA, NAM (10 mM), trichostatin-A (10 μg ml$^{-1}$; Sigma-Aldrich), protease inhibitor cocktail (1×, Roche), olaparib (5 μM; Cayman Chemical), ADP-HPD (250 nM; Millipore Sigma) and benzonase nuclease (0.2 U μl$^{-1}$; Sigma-Aldrich). Where indicated (Fig. 5c), olaparib and ADP-HPD were excluded during the cell lysis step. Samples were probed by western blotting with the following antibodies: rabbit monoclonal anti-PARP1 (Cell Signaling, cat no. 9532, clone 46D11) 1:1000 dilution, rabbit polyclonal (affinity-purified) anti-PAR (Trevigen 4336-APC-050) 1:1000-dilution, rabbit monoclonal anti-histone H4 (Cell Signaling, cat no. 13919, clone D2X4V) 1:1000-dilution, and rabbit polyclonal (affinity-purified) anti-acetyl-histone H4(Lys16) (Millipore Sigma, cat no. 07-329) 1:5000-dilution. Protein bands were visualized with IRDye-labeled secondary antibodies (LI-COR Biosciences) followed by scanning with the Odyssey Digital Infrared Imaging System (LI-COR Biosciences).

**In vivo testing of SBI-797812.** Eight-week-old male C57BL/6 J mice (The Jackson Laboratory) were fed a standard chow diet (Product #2016, Harlan Teklad). All animal studies and procedures were approved by the SBP-Orlando Institutional Animal Care and Use Committee (protocol # 2016-0136). After 1 h of fasting, mice were dosed by i.p. injection of vehicle (10% DMSO, 10% Tween 80 in sterile saline solution) or 20 mg kg$^{-1}$ SBI-797812 (solubilized in vehicle). After 4 h, mice were administered Buthanasia-D (165 mg kg$^{-1}$ body weight; i.p.). Tissues were harvested, flash frozen in liquid nitrogen, lyophilized to dryness and powdered using a Precellys Homogenizer (Bertin Instruments). Tissue powders (15 mg) were subjected to targeted metabolite profiling.

**Quantitation of SBI-797812 in tissue homogenates.** Mouse tissues were harvested 2 h after i.p. dosing of SBI-797812 (20 mg kg$^{-1}$). Tissues were lyophilized and powdered using a Precellys bead-based homogenizer attached to a Crylolys cooling system. Tissue powders (2 mg) were homogenized in 200 μl of 50 mM Tris, pH 7.4 using the Precellys system. Homogenates (25 μl) were extracted with 100 μl ACN. Samples were vortexed and centrifuged at 18,000 × g for 5 min at 10 °C. Supernatants (100 μl) were passed through an AcroPrep Advance 3 K Omega Filter Plate by centrifugation at 3500 × g for 60 min. SBI-797812 in the deproteinized tissue extracts were fractionated using a Dionex Ultimate 3000 UHPLC outfitted with a 2.1 mm × 100 mm, 1.8 μm Cortecs C18 column (Waters Corp.) and run at 55 °C. Samples were injected (2 μl) by an autosampler maintained at 10 °C during the entire run. The mobile phase gradient was 95% A (0.1% formic acid in water) and 5% B (0.1% formic acid in ACN) to 50% A and 50% B over 5.2 min. The gradient began at 5% B (0.7 ml min$^{-1}$ flow rate), was increased from 5 to 10% B (0.7 ml min$^{-1}$ flow rate) from 0.0–5.1 min, and was increased from 10 to 50% B (0.7 ml min$^{-1}$ flow rate) from 5.1–5.2 min. The retention time for SBI-797812 was 4.78 min. The stock solution of SBI-797812 (50 mM) was prepared in DMSO. The working calibration solutions of SBI-797812 (0.01, 0.05, 0.1, 0.5, 1, 5, 10, and 50 μM) were prepared by spiking the DMSO stock solution in 50 mM Tris buffer, pH 7.4. SBI-797812 was quantified with an Agilent 1290 HPLC/6490 triple quadrupole mass spectrometer (Waters Corp.) operated in positive ion mode using electro-spray ionization with an ESI capillary voltage of 3500 V. The electron multiplier voltage was set to 100 V. The ion transfer tube temperature was 325 °C and vaporizer temperature was 325 °C. The ESI source sheath gas flow was set at 10 l min$^{-1}$. The mass spectrometer was operated with a mass resolution of 0.7 Da, cycle

time of 1.9 cycles s$^{-1}$, and nitrogen collision gas pressure was 45 psi for the generation and detection of product ion of SBI-797812. The SRM transition was 403 → 290 and the collision energy to produce the product ion was 25 V for SBI-797812. SBI-797812 raw data was processed using Mass hunter quantitative analysis software (Agilent). The SBI-797812 calibration curve was plotted using the raw area counts from known working calibration solutions.

For details of the Medicinal Chemistry, Mass Spectrometry and Transition Structure Modeling see the Supplementary Methods.

**Statistical analysis.** Statistical analyses (ANOVA or t-test) were performed using GraphPad Prism software. For ANOVA, Dunnett's multiple comparison test or Tukey's multiple comparison tests were performed as indicated in the figure legends. Data was expressed as means ± s.d. $P < 0.05$ was considered statistically significant.

**Reporting summary.** Further information on research design is available in the Nature Research Reporting Summary linked to this article.

## Data availability
The source data underlying Figs. 1c–f, 2a–e, 3a, 3c, 3d, 4a–h, 5a–c, 6a–d and Supplementary Figs. 1–12 are provided as a Source Data file. Any other relevant data is available upon reasonable request from the corresponding authors.

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

## Acknowledgements

We thank the Protein Expression and Purification core (D. Kuijstermans, G. Seo) at SBP in Orlando for producing human NAMPT, mouse NAMPT and human NMNAT1. We are indebted to the SBP Metabolomics core (N.K., J.B., J.C., C.P., and A. Zelenin) for their assistance. Heavy isotope labeled NMN and NAD+ were prepared by D. Divianska and E.H.S. of the CPCCG at SBP in Orlando (partially supported by Florida Translational Research Program, a contract administered by the Florida Department of Health (COHK8)). The Metabolomics core at SBP, which is partnered with the Southeast Center for Integrated Metabolomics (SECIM) was partially supported by NIH grant U24 DK097209 (S.J.G.). Computational support was provided by XSEDE (Extreme Science and Engineering Discovery Environment) and supported by NSF grant number ACI-1548562 (J.S.H.). Funding for this project was provided by SBP, AdventHealth and Daiichi Sankyo Co. Ltd.

## Author contributions

S.R.S. and S.J.G. conceived the project. E.H.S. and C.T-M. conducted the HTS and hit validation. A.B.P. and E.H.S. performed the "hit-to-lead" campaign. S.J.G. designed the studies aimed at characterizing SBI-797812. E.S.B. provided WT cDNA for human NAMPT and purified mutant NAMPT(G217R). M.H., A.K., M.D., R.F., and S.J.G. performed the in vitro, cellular and in vivo studies. C.P., J.C., N.K., and J.B. generated the LC-MS/MS data. S.J.G., A.K., M.D., J.T., S.E.C., and E.S.B. analyzed the enzyme activity data. J.S.H. conducted the computational studies. S.J.G., A.B.P., E.S., and E.S.B. prepared the paper. All authors reviewed the paper and provided comments.
