## [Peer Review File · Nature Communications]

Reviewers' comments:

Reviewer #1 (Remarks to the Author):

This study is an interesting investigation on NAMPT activator. They used elegant chemical synthesis technology to produce SBI-797812, the first evident NAMPT activator.

Several concerns:

1. Due to the structural similarity between SBI-797812 and active-site targeted NAMPT inhibitors, the authors believe that SBI-797812 functions as a competitor to NAMPT inhibitor to combine the crucial site (His247) for NAMPT enzymatic activity. Do these intracellular NAMPT inhibitors include NMN and NAD? In other word, NMN and NAD should also combine the same site His247 with SBI-797812 in NAMPT?

2. An earlier compound P7C3-A20 was claimed to be a direct NAMPT activator. The authors stated that P7C3-A20 is not the direct activator of NAMPT based on their methods. Alternatively, SBI-797812 is the "first-in-class" NAMPT activator. I suggest the authors show several lines of evidence to support their statement. If P7C3-A20 cannot directly boost NMN production, what's the exact action of P3C3-A20 on NAD⁺ biosynthesis? Maybe P7C3-A20 increase NAD via other pathways? More information should be provided to support their conclusion that "SBI-797812 is the first activator of NAMPT".

3. The data illustrated here showed SBI-797812 elevated tissue NAD concentration significantly. Since many disorders such as brain ischemia and obesity/diabetes can be reversed by NAMPT-related substance, such as NMN, NR and NAD. The authors may show some effects of SBI-797812 in disease models, in vivo and in vitro, at least in acute model such as neuronal cells upon ischemic stress. These data are very important for illustrating potential pharmacological value of SBI-797812.

Reviewer #2 (Remarks to the Author):

Gardell S.J. et al.,

Boosting NAD⁺ with a Small Molecule that Activates NAMPT

In this manuscript, the authors conducted a high-throughput screening and identified a novel chemical compound, namely SBI-797812, that can activate nicotinamide phosphoribosyltransferase (NAMPT), a key NAD biosynthetic enzyme. The authors made substantial efforts, carefully characterized the biochemical feature of SBI-797812, and confirmed SBI-797812 increased the production of nicotinamide mononucleotide (NMN) by turning NAMPT into a super catalyst. In addition, they found SBI-797812 treatment increased NAD biosynthesis in human myotubes in vitro and hepatic NAD concentrations in mice in vivo. Given recent enthusiasm in NAD biology research, the topic of this study is very timely and important. Overall, the authors used sophisticated methods and provided comprehensive assessments particularly for the biochemical characterization of SBI-797812. However, the biological and functional significance of this compound might be unclear in the current manuscript.

1) The authors show SBI-797812 treatment significantly increases NAD concentrations in human myotubes (Figure 4d). However, an increase in NAD concentration appears to be very small. Is this small increase sufficient to lead the biological effects in human myotubes? It would be important to evaluate the effects of SBI-797812 on putative downstream targets of NAMPT-NAD in myotubes. For example, it is well known NAD-dependent protein deacetylase SIRT1 regulates key cellular metabolic pathways, such as mitochondrial biology, oxidative metabolism, and fatty acid oxidation. Therefore, it is recommended the authors evaluate these metabolic pathways.

2) In Figure 4e, the authors found intraperitoneal administration of SBI-79812 significantly increased NAD concentrations in liver. Accumulating evidence suggests NAD (and its downstream mediators) plays a critical role in regulating liver metabolism, such as gluconeogenesis, inflammation, and mitochondrial biology. Does this compound affect these cellular metabolic pathways and/or whole-body glucose metabolism (e.g. blood glucose, insulin, glucose tolerance)? These biological and functional assessments would increase the impact of the discovery and uniqueness of this compound.

3) The authors may want to measure tissue concentrations of this compound in SBI-797812 treated mice. This information would help understand why SBI-797812 increases NAD concentrations only in liver but not other organs, in conjunction with the blood pharmacokinetics of this compound (Figure S9b).

4) The bioavailability of oral SBI-797812 administration appears to be very low (Figure S9a). The reviewer thinks these findings could make it difficult to use this compound in humans unless the oral bioavailability is very different between humans and mice. In addition, other NAD boosters, such as

NMN and nicotinamide riboside (NR), provide excellent bioavailability (via oral administration) and translational and clinical potential. It would be important to discuss if and how this new compound can be complimentary or superior to other NAD boosters in terms of application in humans.

5) ANOVA, but not t-test, should be more appropriate to compare values among multiple (>2) groups (e.g. Figures 1c, 1e, 1f, 2e, 3c, 4a, 4d).

Reviewer #3 (Remarks to the Author):

The manuscript by Gardell, et al., entitled “Boosting NAD⁺ with a small-molecule that activates NAMPT” (NCOMMS-18-34344-T) describes the ability of a small-molecule (SBI-797812) to increase NAMPT-mediated production of NMN (an immediate biological precursor of NAD). The described phenomenon is scientifically intriguing and such “NAMPT activators” are of high interest in the neurodegeneration field. A prior publication exists that details similar NAMPT activating properties associated with the P7C3 class of compounds (Cell 2014, 158, 1324; see additional comments regarding P7C3 and its analogs below). However, these molecules are structurally-unrelated to the activators contained in the current work. The ability to transform potent NAMPT inhibitors such as GNI-50 into NAMPT activators by moving the location of a key pyridine N-atom is also surprising and novel.

I am concerned, however, that the activating effects described in the current manuscript may result primarily from the chosen *in vitro* assay systems and/or non-specific effects and may not meaningfully translate into functional *in vitro* and/or *in vivo* applications. My reasons for this concern are provided below along with several other suggestions to augment the manuscript. I leave it to the editor to decide how best to proceed regarding these items. Publication of the work in its present form may spur additional research in the NAMPT activator field that may help establish the functional relevance I currently question. Alternatively, including data demonstrating such translation (as well as the other additions suggested below) in a revised version of the manuscript would solidly connect NAMPT activation with desired functional impacts.

1. The SBI-797812 compound is typically employed in the described enzyme and cell-based assays at relatively high concentrations (>1 μ M) in order to produce the NAMPT activating effects. I am curious to know how these concentrations compare to the affinity of the compound for the NAMPT enzyme. Biophysical methods were previously used to assess the affinity of various molecules for NAMPT (for one such example, see: J. Med. Chem. 2014, 57, 770). Related assessments should be

used to quantitatively measure the NAMPT affinity of SBI-797812. If such affinity is significantly more potent than the μM SBI-797812 concentrations frequently utilized in the current manuscript, I worry that non-specific interactions with NAMPT (or some other biological entity) may be associated with the activation events. In this case, the authors should comment regarding why the relatively high SBI-797812 concentrations are required.

2. A co-crystal structure of SBI-797812 in complex with NAMPT should be obtained. Such a structure would confirm specific association of the molecule with the enzyme and would potentially help explain the mechanism of NAMPT activation. Many co-crystal structure examples of related molecules in complex with NAMPT exist in the chemical literature. These include compounds with relatively weak binding affinities (see: *J. Med. Chem.* 2014, 57, 770 and *Bioorg. Med. Chem. Lett.* 2014, 24, 954 for some examples) Measuring the affinity of SBI-797812 for NAMPT (suggested in item #1 above) should also assist co-crystallization efforts.

3. The manuscript describes *in vitro* experiments using A549 cells that measure production of NAD. A549 cells are known to be proficient in a second biochemical pathway that can produce NAD from nicotinic acid (NA) without the involvement of NAMPT (see: *Neoplasia* 2013, 15, 1151). Were NA levels in the cell media properly controlled to ensure that this second pathway did not influence the NAD outcomes? Alternatively, can the cell-based experiments be conducted in an alternate line that lacks the second NAD-producing biochemical pathway?

4. In the beginning of the Discussion section, the authors mention that they failed to confirm the NAMPT activating properties of P7C3. This is a very important revelation as P7C3 and related molecules provide a crucial link between NAMPT activation and functional neuroprotective effects. The following questions should be addressed by the authors of the current manuscript.

A. How do the NAMPT activation assay conditions differ between the current manuscript and those reported for characterization of the P7C3 class of molecules (*Cell* 2014, 158, 1324)?

B. If the assay conditions are similar, how do the authors explain the lack of P7C3 NAMPT activation activity? Do such similarities and lack of P7C3 activity imply the potential for assay variations to influence apparent activation outcomes? If so, should the results for SBI-797812 be viewed with caution?

C. Were other P7C3 analogs tested by the current manuscript authors for NAMPT activation? For example, P7C3-A20 is commercially available and is reported to afford dose-dependent NAMPT activation effects (Cell 2014, 158, 1324; Figure 7A). How does it fare in the NAMPT activation assays described in the current manuscript (see item 4B above)?

D. Does SBI-797812 exhibit neuroprotective effects in cell culture experiments? The Cell 2014, 158, 1324 paper describes one such assessment that could possibly be employed. This question is related to the concern mentioned at the beginning of this review about the functional relevance of the described NAMPT activation findings.

5. In preparation for mouse in vivo studies, the authors describe efforts to characterize SBI-797812 using mouse NAMPT enzyme (page 9). In the case of NAMPT inhibitors, however, cross-species profiling using NAMPT enzymes did not fully reveal significant activity differences that manifested in cells and in vivo (see: Bioorg. Med. Chem. Lett. 2013, 23, 5488). Thus, the NAMPT activating properties of SBI-797812 should be assessed using mouse cells in addition to the mouse NAMPT enzyme.

6. Some mouse in vivo data associated with SBI-797812 are included in the manuscript. However, it is clear from the associated PK profile (Figure S9) that the molecule is a poor in vivo tool compound. Accordingly, an alternate NAMPT activator should be identified with significantly improved in vivo properties relative to SBI-797812 for use in the animal studies. Also, can the in vivo NAD levels be measured in a second location that is distinct from the liver? The Bioorg. Med. Chem. Lett. 2013, 23, 5488 paper describes NAD measurements in blood.

7. Near the bottom of page 7, the authors speculate that NAMPT-bound SBI-797812 “exerts a water shield” that enables pHis247 to react with nucleophiles other than water. Why does this alternate reactivity lead to activation of the NAMPT enzyme?

Reviewer #1 (Remarks to the Author):

This study is an interesting investigation on NAMPT activator. They used elegant chemical synthesis technology to produce SBI-797812, the first evident NAMPT activator.

Several concerns:

1. Due to the structural similarity between SBI-797812 and active-site targeted NAMPT inhibitors, the authors believe that SBI-797812 functions as a competitor to NAMPT inhibitor to combine the crucial site (His247) for NAMPT enzymatic activity. Do these intracellular NAMPT inhibitors include NMN and NAD? In other word, NMN and NAD should also combine the same site His247 with SBI-797812 in NAMPT?

Author response: It was shown previously that NAD⁺ inhibits NAMPT activity (mentioned on Page 8 of the manuscript). NMN is not considered to be a NAMPT inhibitor; however, NMN will impede NAMPT-mediated NMN production by fueling NAMPT's "reverse reaction". The ability of SBI-797812 to blunt the inhibitory effect of NAD⁺ (see Fig. 2e and summary Fig. 6) suggests mutually-exclusive binding of SBI-797812 and NAD⁺. NMN and SBI-797812 also likely have overlapping binding sites based on their shared structural feature, a pyridine ring. Since the binding of SBI-797812 and FK-866 are mutually-exclusive, as is the binding of NMN and FK-866, we infer that the binding of NMN and SBI-797812 would also be mutually exclusive. The role of His247 in these binding interactions is not known. One possibility being explored is that SBI-797812 and these other ligands do not bind simultaneously to the same active site in the NAMPT homodimer due to subunit non-equivalence arising from catalytic cooperativity of the two active sites in the NAMPT homodimer.

2. An earlier compound P7C3-A20 was claimed to be a direct NAMPT activator. The authors stated that P7C3-A20 is not the direct activator of NAMPT based on their methods. Alternatively, SBI-797812 is the "first-in-class" NAMPT activator. I suggest the authors show several lines of evidence to support their statement.

Author response: We have included data in the manuscript showing that P7C3-S243 (close structural analog of P7C3-A20) does not bind to NAMPT (PTS assay; Supplementary Figure 3a). Moreover, P7C3-A2 or P7C3-S243 did not activate NAMPT using the "gold standard" LC-MS/MS assay of NMN production by purified NAMPT (Supplementary Figure 3b).

If P7C3-A20 cannot directly boost NMN production, what's the exact action of P3C3-A20 on NAD⁺ biosynthesis?

Author response: This question is tangential to our primary research focus (direct NAMPT activators) and, hence, we cannot shed light on the answer.

Maybe P7C3-A20 increase NAD via other pathways?

Author Response: It is possible that P7C3 might increase NAD via a pathway other than direct NAMPT activation. However, deciphering the pharmacological effects of P7C3

(beyond excluding it as a direct NAMPT activator) was not a focus of our research efforts.

More information should be provided to support their conclusion that “SBI-797812 is the first activator of NAMPT”.

Author response: We have incorporated the requested information showing that 2 members of the P7C3 chemical series (disclosed by the McKnight group) are not direct NAMPT activators (see Supplementary Figure 3).

3. The data illustrated here showed SBI-797812 elevated tissue NAD concentration significantly. Since many disorders such as brain ischemia and obesity/diabetes can be reversed by NAMPT-related substance, such as NMN, NR and NAD. The authors may show some effects of SBI-797812 in disease models, in vivo and in vitro, at least in acute model such as neuronal cells upon ischemic stress. These data are very important for illustrating potential pharmacological value of SBI-797812.

Authors response: We agree that this is a crucial research objective and such studies are ongoing. The thrust of the present manuscript was to disclose SBI-797812, a first-in-class direct NAMPT activator. However, in response to the reviewer, we now show data that examines the impact of SBI-797812 on sirtuin (Fig. 4f) and PARP (Fig. 4g) activity in A549 cells. The pharmacological effects of SBI-797812 analogs with superior properties are currently being determined.

Reviewer #2 (Remarks to the Author):

Gardell S.J. et al.,

Boosting NAD⁺ with a Small Molecule that Activates NAMPT

In this manuscript, the authors conducted a high-throughput screening and identified a novel chemical compound, namely SBI-797812, that can activate nicotinamide phosphoribosyltransferase (NAMPT), a key NAD biosynthetic enzyme. The authors made substantial efforts, carefully characterized the biochemical feature of SBI-797812, and confirmed SBI-797812 increased the production of nicotinamide mononucleotide (NMN) by turning NAMPT into a super catalyst. In addition, they found SBI-797812 treatment increased NAD biosynthesis in human myotubes in vitro and hepatic NAD concentrations in mice in vivo. Given recent enthusiasm in NAD biology research, the topic of this study is very timely and important. Overall, the authors used sophisticated methods and provided comprehensive assessments particularly for the biochemical characterization of SBI-797812. However, the biological and functional significance of this compound might be unclear in the current manuscript.

1) The authors show SBI-797812 treatment significantly increases NAD concentrations in human myotubes (Figure 4d). However, an increase in NAD concentration appears to be very small. Is this small increase sufficient to lead the biological effects in human myotubes?

Authors response: Our current data do not allow us to answer this excellent question. This matter is currently being addressed in human myotubes with superior compounds in our collection and various tests for their potential impact on cellular metabolism. It is

crucial to emphasize a point stated in the Results (page 9) and Discussion (page 13). Homeostatic mechanisms appear to exist that place a ceiling effect on the steady state level of NAD⁺. Hence, simply assaying the levels of NAD⁺ will likely underestimate the impact of the NAMPT activator on NAD⁺ biosynthetic flux. We strongly believe that elevated NAD⁺ biosynthetic flux in the presence of SBI-797812 (as shown for A549 cells in Fig. 4c) will be the key driver of downstream biological effects. Exploring this hypothesis using highly-relevant cell line such as myotubes is a research priority

- It would be important to evaluate the effects of SBI-797812 on putative downstream targets of NAMPT-NAD in myotubes. For example, it is well known NAD-dependent protein deacetylase SIRT1 regulates key cellular metabolic pathways, such as mitochondrial biology, oxidative metabolism, and fatty acid oxidation. Therefore, it is recommended the authors evaluate these metabolic pathways.

Authors response: We now disclose data showing the impact of SBI-797812 on SIRT1 activity (Fig. 4f) and PARP activity (Fig. 4g) in A549 cells. We focused on A549 cells because these cells exhibited a more robust NAD⁺-elevating response to SBI-797812. However, studies with myotubes are underway to more deeply explore the impact of SBI-797812 (and its analogs) in this cell type with obvious therapeutic implications.

2) In Figure 4e, the authors found intraperitoneal administration of SBI-79812 significantly increased NAD concentrations in liver. Accumulating evidence suggests NAD (and its downstream mediators) plays a critical role in regulating liver metabolism, such as gluconeogenesis, inflammation, and mitochondrial biology. Does this compound affect these cellular metabolic pathways and/or whole-body glucose metabolism (e.g. blood glucose, insulin, glucose tolerance)?

Authors response: Deciphering the pharmacological effects of SBI-797812 that arise from increased NAMPT activity is an important research goal. It is the focus of ongoing investigations and will be disclosed in a future publication when using an SBI-797812 analog that has more favorable ADME properties which will enable chronic dosing studies (yielding sustained compound exposure).

These biological and functional assessments would increase the impact of the discovery and uniqueness of this compound.

Authors response: “This” compound (i.e., SBI-797812) is not suitable for the biological and functional assessments that are cited by the reviewer due to its pharmacokinetics shortcomings. However, we look forward to sharing such data that is currently being generated with a more “drug-like” derivative of SBI-7697812.

3) The authors may want to measure tissue concentrations of this compound in SBI-797812 treated mice. This information would help understand why SBI-797812 increases NAD concentrations only in liver but not other organs, in conjunction with the blood pharmacokinetics of this compound (Figure S9b).

Authors response: We have performed this experiment using tissue powders collected from previous *in vivo* studies with SBI-797812 (see Supplementary Figure 11). Note that the tracer data with ¹³C-NAM (Liu et al which is cited in the manuscript and our unpublished data) revealed that profound differences exist across individual tissues with respect to NAMPT flux. Hence, the intracellular NAD⁺ levels reflect the balance between synthesis and degradation. We believe that inter-tissue differences in NAD⁺

consumption of *de novo* generated NAD⁺ is another important driver of the disparate effects of the NAMPT activator in mice. Studies are underway to explore this likely possibility using a compound that has more favorable pharmacokinetic properties.

4) The bioavailability of oral SBI-797812 administration appears to be very low (Figure S9a). The reviewer thinks these findings could make it difficult to use this compound in humans unless the oral bioavailability is very different between humans and mice.

Authors response: We agree with the reviewer. We have produced superior compounds from the perspective of both potency and pharmacokinetics that are more attractive clinical candidates. They are the focus of the drug development campaign.

In addition, other NAD boosters, such as NMN and nicotinamide riboside (NR), provide excellent bioavailability (via oral administration) and translational and clinical potential. It would be important to discuss if and how this new compound can be complimentary or superior to other NAD boosters in terms of application in humans.

Authors response: We now include the following statement on page 13 of the Discussion:

Comparing the therapeutic utility of small molecule NAMPT activators to NAD⁺ boosters such as nicotinamide riboside (NR) or NMN will be essential. The fact that SBI-797812 acts “catalytically” to promote NAD⁺ synthesis along with its ability to suppress feedback inhibition of NAMPT activity by NAD⁺ (Fig. 2e) are two discriminating attributes that are likely to be advantageous.

5) ANOVA, but not t-test, should be more appropriate to compare values among multiple (>2) groups (e.g. Figures 1c, 1e, 1f, 2e, 3c, 4a, 4d).

Authors response: We thank the reviewer for pointing out this oversight. The statistical analysis has been modified as suggested.

Reviewer #3 (Remarks to the Author):

The manuscript by Gardell, et al., entitled “Boosting NAD⁺ with a small-molecule that activates NAMPT” (NCOMMS-18-34344-T) describes the ability of a small-molecule (SBI-797812) to increase NAMPT-mediated production of NMN (an immediate biological precursor of NAD). The described phenomenon is scientifically intriguing and such “NAMPT activators” are of high interest in the neurodegeneration field. A prior publication exists that details similar NAMPT activating properties associated with the P7C3 class of compounds (Cell 2014, 158, 1324; see additional comments regarding P7C3 and its analogs below). However, these molecules are structurally-unrelated to the activators contained in the current work. The ability to transform potent NAMPT inhibitors such as GNI-50 into NAMPT activators by moving the location of a key pyridine N-atom is also surprising and novel.

I am concerned, however, that the activating effects described in the current manuscript may result primarily from the chosen in vitro assay systems and/or non-specific effects and may not meaningfully

translate into functional in vitro and/or in vivo applications. My reasons for this concern are provided below along with several other suggestions to augment the manuscript. I leave it to the editor to decide how best to proceed regarding these items. Publication of the work in its present form may spur additional research in the NAMPT activator field that may help establish the functional relevance I currently question. Alternatively, including data demonstrating such translation (as well as the other additions suggested below) in a revised version of the manuscript would solidly connect NAMPT activation with desired functional impacts.

1. The SBI-797812 compound is typically employed in the described enzyme and cell-based assays at relatively high concentrations (>1 μM) in order to produce the NAMPT activating effects. I am curious to know how these concentrations compare to the affinity of the compound for the NAMPT enzyme. Biophysical methods were previously used to assess the affinity of various molecules for NAMPT (for one such example, see: J. Med. Chem. 2014, 57, 770). Related assessments should be used to quantitatively measure the NAMPT affinity of SBI-797812. If such affinity is significantly more potent than the μM SBI-797812 concentrations frequently utilized in the current manuscript, I worry that non-specific interactions with NAMPT (or some other biological entity) may be associated with the activation events. In this case, the authors should comment regarding why the relatively high SBI-797812 concentrations are required.

Authors response: The EC_{50} value of SBI-797812 (estimate for its binding affinity) was 0.37 μM , see page 4. Hence, using SBI-797812 concentrations that are >1 μM to saturate the enzyme is consistent with standard practice. The reviewer appears to be referring to the higher concentrations of SBI-797812 that are used to see an effect in cultured cells. The explanation for the apparent lower potency of the compound can be attributed to protein binding of the compound by serum (e.g., 10% FBS in our culture medium) and intracellular proteins. This is a very common finding. We have included a sentence in the manuscript that addresses this point: “The apparent decreased potency of SBI-797812 in the cellular assays likely reflects binding by intracellular proteins and serum-containing cell culture media” (see page 8).

2. A co-crystal structure of SBI-797812 in complex with NAMPT should be obtained. Such a structure would confirm specific association of the molecule with the enzyme and would potentially help explain the mechanism of NAMPT activation. Many co-crystal structure examples of related molecules in complex with NAMPT exist in the chemical literature. These include compounds with relatively weak binding affinities (see: J. Med. Chem. 2014, 57, 770 and Bioorg. Med. Chem. Lett. 2014, 24, 954 for some examples) Measuring the affinity of SBI-797812 for NAMPT (suggested in item #1 above) should also assist co-crystallization efforts.

Authors response: This is a crucial objective that is currently being pursued and will be disclosed in a future publication.

3. The manuscript describes in vitro experiments using A549 cells that measure production of NAD. A549 cells are known to be proficient in a second biochemical pathway that can produce NAD from nicotinic acid (NA) without the involvement of NAMPT (see: Neoplasia 2013, 15, 1151). Were NA levels in the cell media properly controlled to ensure that this second pathway did not influence the NAD outcomes?

Authors response: The cell media used for these experiments was prepared from large batches which mitigates any concerns about inter-experimental variation of the NA level (concerns about NA as a confounder are also addressed in the next Q&A).

Alternatively, can the cell-based experiments be conducted in an alternate line that lacks the second NAD-producing biochemical pathway?

Authors response: Yes – but we believe such a study is unnecessary. Treatment of the A549 cells with FK-866 abolished NAD⁺ production (see Fig. 4c). This definitive result confirms that NAMPT in these cells and under our standard assay conditions plays the exclusive role in mediating NAD⁺ synthesis. This conclusion is further supported by our inability to detect nicotinic acid mononucleotide (NAMN) or nicotinic acid dinucleotide (NAAD) in the A549 cells (+/- SBI-797812) using mass spectrometry.

4. In the beginning of the Discussion section, the authors mention that they failed to confirm the NAMPT activating properties of P7C3. This is a very important revelation as P7C3 and related molecules provide a crucial link between NAMPT activation and functional neuroprotective effects. The following questions should be addressed by the authors of the current manuscript.

A. How do the NAMPT activation assay conditions differ between the current manuscript and those reported for characterization of the P7C3 class of molecules (Cell 2014, 158, 1324)?

Authors response: There are profound differences between the *in vitro* assay used by the McKnight group (Wang *et al*, Cell 158, 1324-1334 (2014) and the *in vitro* assay(s) that we employed in our investigation. Wang *et al* deduced activation of NAMPT activity using a coupled enzymatic assay in which 3 enzymes were present: NAMPT, NMNAT, alcohol dehydrogenase (formation of NADH final product). Our assessment of the activating effect of P7C3 on NAMPT activity measured direct production of NMN (presence of NAMPT only) using a fluorescent assay or mass spectrometric analysis (with excellent agreement between these orthogonal assays). The more complex “coupled reaction” performed by Wang *et al* is subject to “non-NAMPT-targeted effects” by the compound giving rise to spurious effects that might be erroneously attributed to NAMPT modulation. Our direct assay provides a definitive functional assessment of direct target engagement with NAMPT.

McKnight lab NAMPT assay components (Coupled NAMPT assay; monitor for putative NAD signal

- 50 mM Tris (pH 8.0)
- 0.4 mM PRPP
- 150 μM nicotinamide
- 2.5 mM ATP
- 12 mM MgCl₂
- 1.5% (V/V)(ethanol
- 10 mM semicarbazide
- 0.02% BSA
- 2.4 μg/ml NMNAT
- 0.4 U alcohol dehydrogenase

- 1 μ M NAMPT

Gardell lab NAMPT assay components (Direct NAMPT assay; monitor for NMN production)

- 50 mM Tris (pH 7.5)
- 50 μ M PRPP
- 25 μ M nicotinamide
- 2.0 mM ATP
- 2 mM DTT
- 10 mM $MgCl_2$
- 30 nM NAMPT

B. If the assay conditions are similar, how do the authors explain the lack of P7C3 NAMPT activation activity?

Authors response: The assay conditions are not similar (coupled vs. direct assay as described above).

Do such similarities and lack of P7C3 activity imply the potential for assay variations to influence apparent activation outcomes? If so, should the results for SBI-797812 be viewed with caution?

Authors response: The assays are not similar. We used an LC-MS/MS assay for NMN (with NMN internal standard) which is the “gold standard” for measuring NMN. The “similarities” between P7C3 and SBI-797812 only relate to the direct NAMPT activator claim in the Cell paper by the McKnight group. Our rigorous investigation of SBI-797812 yielded data that was highly reproducible. Moreover, the structure-activity relationship (SAR) of the SBI-797812 chemical series is highly-predictive as gleaned from analysis of hundreds of compound derivatives (unpublished data, manuscript in preparation).

The data presented in this manuscript convincingly show that SBI-797812 is a direct NAMPT activator. We look forward to disclosing the results from the extensive lead optimization campaign that has produced numerous compounds with substantially greater activating potency. We emphasize that SBI-797812 is not a “one-off” success story but is a prototype for a novel class of NAMPT activators.

C. Were other P7C3 analogs tested by the current manuscript authors for NAMPT activation?

Authors response: Yes. In response to the reviewer’s comment, we now present data with P7C3-A20 in addition to P7C3-S243 (see Supplementary Figure 3b).

For example, P7C3-A20 is commercially available and is reported to afford dose-dependent NAMPT activation effects (Cell 2014, 158, 1324; Figure 7A). How does it fare in the NAMPT activation assays described in the current manuscript (see item 4B above)?

Authors response: : P7C32-A20 did not promote NMN production by NAMPT using our NAMPT activation assay

D. Does SBI-797812 exhibit neuroprotective effects in cell culture experiments?

Authors response: We have not yet done this experiment. However, the potential utility of our compounds for neuroprotection is an exciting possibility. Hence, experiments such as suggested by the reviewer will be performed but with a superior analog of SBI-797812.

The Cell 2014, 158, 1324 paper describes one such assessment that could possibly be employed. This question is related to the concern mentioned at the beginning of this review about the functional relevance of the described NAMPT activation findings.

5. In preparation for mouse in vivo studies, the authors describe efforts to characterize SBI-797812 using mouse NAMPT enzyme (page 9). In the case of NAMPT inhibitors, however, cross-species profiling using NAMPT enzymes did not fully reveal significant activity differences that manifested in cells and in vivo (see: Bioorg. Med. Chem. Lett. 2013, 23, 5488). Thus, the NAMPT activating properties of SBI-797812 should be assessed using mouse cells in addition to the mouse NAMPT enzyme.

Authors response: We thank the reviewer for this suggestion. In response, the impact of SBI-797812 was assessed with mouse primary myotubes (see Fig. 4e). We show that SBI-797812 increased both NMN and NAD⁺ in the mouse myotubes.

6. Some mouse in vivo data associated with SBI-797812 are included in the manuscript. However, it is clear from the associated PK profile (Figure S9) that the molecule is a poor in vivo tool compound. Accordingly, an alternate NAMPT activator should be identified with significantly improved in vivo properties relative to SBI-797812 for use in the animal studies.

Authors response: We strongly agree with the reviewer. The description of such compounds with improved “in vivo properties” are beyond the scope of our current manuscript. We described the “first-in-class” (prototypical) direct NAMPT activator within this communication. Such compounds with superior ADME properties have been identified and will be the subject of future publications

Also, can the in vivo NAD levels be measured in a second location that is distinct from the liver? The Bioorg. Med. Chem. Lett. 2013, 23, 5488 paper describes NAD measurements in blood.

Authors response: Yes. Indeed, we reported the levels of NAD in heart, gastrocnemius and quadriceps skeletal muscle (see Fig 5).

7. Near the bottom of page 7, the authors speculate that NAMPT-bound SBI-797812 “exerts a water shield” that enables pHis247 to react with nucleophiles other than water. Why does this alternate reactivity lead to activation of the NAMPT enzyme?

Authors response: With its uncommon autophosphorylation at active site H247, HisNAMPT is a unique catalyst among the ribosyltransferase family. Through ATP-consumption and an enhanced affinity for PRPP, NAM is efficiently captured and recycled by the enzyme, thus making the NMN synthetic reaction virtually irreversible. Previous reports claimed the instability of phosphorylated NAMPT, in part due to nucleophilic attack by water at p-H247; so that high ATP concentrations were required to fully saturate the catalyst. Supporting the role of SBI-797812 as a direct NAMPT activator, our data established the tighter ATP affinity for the enzyme target ($K_m = 0.29$ mM with

SBI-797812, versus 1.73 mM). Furthermore, we identified an unprecedented pathway in which pyrophosphate was catabolized into triphosphate via a S_N2 reaction with phosphorylated NAMPT (such pathway is reminiscent of the Ap₄ formation also catalyzed by NAMPT; Amici, A. *et al* Cell Chem Biol 2017 and our supplementary data). By promoting such catabolic reaction, SBI activator is rigging the Thermodynamic Scheme (Le Chatelier's Principle).

REVIEWERS' COMMENTS:

Reviewer #1 (Remarks to the Author):

I have no more questions for the part of my comments on the manuscript.
Chao-Yu Miao

Reviewer #2 (Remarks to the Author):

Gardell S.J. et al.,

Boosting NAD⁺ with a Small Molecule that Activates NAMPT

The authors made efforts, added new results, and improved the manuscript. The reviewer agrees that the steady levels of NAD may not reflect the actual impact of the activators. This is why the reviewer thinks it is important to evaluate the effects of the activators on any specific biological and physiological outcome (as the reviewer #3 also pointed out). New data (SIRT1 and PARP activities) are definitely important additions in this manuscript but may not be directly associated with any functional aspects. Although biochemical characterization of the new activators is novel and potentially important in the field, lack of *in vitro* and *in vivo* functional assays somewhat reduces the overall impact of this study.

Reviewer #3 (Remarks to the Author):

I think the authors did a good job responding to all the reviewer questions regarding the technical aspects of the NAMPT activation assessments. The concerns raised by all three reviewers regarding translation of the described data into functional *in vitro* and/or *in vivo* applications remain. However, I am comfortable proceeding with publication of the work in its current form.